# Bipolar cell targeted optogenetic gene therapy restores parallel retinal signaling and high-level vision in the degenerated retina

Jakub Kralik [1,3], Michiel van Wyk[1,3], Nino Stocker[1,2] & Sonja Kleinlogel [1✉]

Optogenetic gene therapies to restore vision are in clinical trials. Whilst current clinical approaches target the ganglion cells, the output neurons of the retina, new molecular tools enable efficient targeting of the first order retinal interneurons, the bipolar cells, with the potential to restore a higher quality of vision. Here we investigate retinal signaling and behavioral vision in blind mice treated with bipolar cell targeted optogenetic gene therapies. All tested tools, including medium-wave opsin, Opto-mGluR6, and two new melanopsin based chimeras restored visual acuity and contrast sensitivity. The best performing opsin was a melanopsin-mGluR6 chimera, which in some cases restored visual acuities and contrast sensitivities that match wild-type animals. Light responses from the ganglion cells were robust with diverse receptive-field types, inferring elaborate inner retinal signaling. Our results highlight the potential of bipolar cell targeted optogenetics to recover high-level vision in human patients with end-stage retinal degenerations.

[1] Institute of Physiology and Department for BioMedical Research (DBMR), University of Bern, Bern, Switzerland. [2] Present address: Swiss Institute of Allergy and Asthma Research (SIAF), Davos, Switzerland. [3] These authors contributed equally: Jakub Kralik, Michiel van Wyk. ✉email: sonja.kleinlogel@unibe.ch

Optogenetic vision restoration has progressed rapidly to clinical trials. As opposed to other tissues, the retina is naturally accessible to light, and the eye is partially immune-privileged[1], reducing the risk of adverse effects. In contrast to replacement gene therapies, optogenetics is a mutation-independent one-for-all therapy for photoreceptor degenerative diseases that renders remaining inner retinal cells light sensitive. The sole prerequisite is surviving inner retinal neurons. It was shown that the inner retina, in particular the bipolar cells, remain intact for years after photoreceptor loss[2], and seminal optogenetic restoration studies have demonstrated their functional integrity[3–6].

Current optogenetic trials (ClinicalTrials.gov Identifier: NCT03326336, NCT02556736, NCT04278131) introduce channelrhodopsin (ChR) variants to the retinal ganglion cells (RGCs), the output neurons of the retina, turning them into direct light detectors. However, a substantial amount of image processing is performed presynaptic to the RGCs, by the bipolar and amacrine cell networks. The inner retinal circuitry dissects the visual scene into ~30 parallel channels that encode, for example, luminance, local contrast, and directed movement[7]. This parallel processing of visual information is by-passed when directly imparting light sensitivity to the RGCs.

As photoreceptors degenerate, the bipolar cells, which normally receive input from the photoreceptors, become the first surviving cells within the retinal hierarchy. Direct optogenetic stimulation of bipolar cells, therefore, preserves inner retinal processing, and with that, diverse RGC receptive-fields, which has the potential to restore higher quality vision compared to RGC targeted approaches.

Recently developed synthetic AAVs and ON-bipolar cell (OBC) specific promoters have paved the way for OBC-targeted optogenetic vision restoration[8–10]. To date, both native and engineered opsins have been expressed in OBCs to successfully restore retinal light responses[3–6,8,11]. When ectopically expressed in OBCs, these opsins trigger the Gαo signaling cascade of the primary OBC receptor, the metabotropic glutamate receptor 6 (mGluR6). mGluR6 is exclusively expressed in the postsynaptic region of OBCs[12] and signals via the Gαo G-protein pathway[13] to gate transient receptor potential melastatin 1 (TRPM1) non-selective cation channels[14,15]. A key advantage of this approach is that signal amplification inherent to metabotropic receptors make GPCR opsins approximately 1000-fold more light sensitive compared to ChRs. Rhodopsin[3] and medium-wave cone opsin (OPN1MW)[5,8] have both been successfully expressed in OBCs to restore vision. Both these opsins naturally activate Gt (transducin) in photoreceptors, but also activate Gαo, which belongs to the same G-protein family (Gi/o). In an attempt to optimize activation of the OBC's Gαo signaling pathway, we have previously engineered a chimera of melanopsin with the intracellular domains exchanged by those of mGluR6 (Opto-mGluR6). This successfully converted the Gαq-protein tropism of melanopsin to the Gαo tropism of mGluR6[6].

Here we elucidate the full potential of OBC-targeted Opto-GPCRs in vision restoration. We functionally compare Opto-mGluR6[6], OPN1MW-mGluR6[8], and two new melanopsin-mGluR6 chimeras. We show that all opsins are functional and able to restore vision at the retinal, cortical, and behavioral levels. The C-terminus of mGluR6 enhanced functional opsin expression. Signaling in the RGC population was diverse with ON and OFF as well as transient and sustained visual channels and had the capacity to adapt to environmental light intensities. Pathophysiological changes associated with retinal degeneration introduced some "sluggishness" to restored RGC light responses. Nonetheless, we show that an OBC-targeted optogenetic therapy has the potential to restore visual acuities and contrast sensitivities in blind rd1 mice close to wild-type values. On the background of a recent report of successful object

localization by a patient treated with the ChR variant ChrimsonR targeted at the macular RGCs[16], the encouraging results of this study anticipate that higher quality vision can be restored in human patients through OBC-targeted optogenetic gene therapies.

## Results

**Screening of opsin constructs in HEK293-GIRK cells.** GPCR opsins previously used in OBC-targeted gene therapies include the Opto-mGluR6 chimera[6], rhodopsin[3], medium-wave cone opsin (OPN1MW)[5,8] and lamprey parapinopsin[17]. Our aim was to examine the maximum potential of this therapeutic approach. In light of potential clinical use we initially focused on human OPN1MW (a canonical bleachable ciliary pigment) and human melanopsin (a bleach resistant tri-stable opsin)[18]. We tested Opto-mGluR6[6], two additional melanopsin-mGluR6 chimeras, Mela(CTmGluR6) and Mela(CT+IL3mGluR6), and OPN1MW(CTmGluR6)[8]. In Mela(CTmGluR6) and in OPN1MW(CTmGluR6), we replaced the native opsin carboxyl termini by that of mGluR6, the resident receptor in OBCs that is naturally activated by glutamate released from the photoreceptors[19]. This was done to accelerate the kinetics of melanopsin[20,21] and to support subcellular trafficking in the target OBCs[6]. Mela(CT+IL3mGluR6) has, in addition, the intracellular loop 3 (IL3) of melanopsin replaced by that of mGluR6, a region known to be important for G-protein selectivity[22]. Mela(CT+IL3mGluR6) was not straightforward to design, since IL3 is particularly long in melanopsin but very short in mGluR6. Unlike the initial design of Opto-mGluR6 where the entire IL3 of melanopsin was replaced, we here opted for a strategy where the relatively short IL3 of mGluR6 replaced only part of melanopsin's IL3.

We first tested the expression and function of all chimeric constructs in HEK293-GIRK cells, which stably express the Kir3.1/Kir3.2 potassium channel opened by Gβγ G-protein subunits released by activated G-proteins of the Gαi/o family (Supplementary Fig. 1a). Since the TRPM1 channels driving excitation in OBCs are also Gβγ-gated (Supplementary Fig. 1b)[23], GIRK channel coupling presents a relevant in vitro functional assay for the target mGluR6 pathway in OBCs. While functional recordings were made using opsin-IRES-TurboFP635 constructs, we also fused all opsin chimeras directly to the fluorescent protein mKate2 to monitor membrane localization. All proteins targeted well to the cell membrane, and expression extended into the fine filipods (Fig. 1a). All opsin constructs activated GIRK currents in response to a constant 470 nm light stimulus (5 s, $10^{14}$ photons/cm$^2$/s, Fig. 1b, c). Currents elicited by native melanopsin were significantly smaller than those elicited by Mela(CTmGluR6) ($p = 0.0005$), Mela(IL3+CTmGluR6) ($p = 0.0004$), OPN1MW ($p = 0.026$) and OPN1MW(CTmGluR6) ($p = 0.0001$). However, the currents elicited by Opto-mGluR6 ($3.9 \pm 1.8$ pA/pF, $n = 4$; mean ± SEM) were significantly smaller than those elicited by native melanopsin ($21.9 \pm 5.7$ pA/pF, $n = 4$; $p = 0.021$) and all other constructs tested ($p < 0.0001$). This may be due to the extensive chimeric design, which may reduce the efficacy of the engineered Opto-mGluR6. Alternatively, Opto-mGluR6's tropism may be almost completely shifted towards Gαo, which is not expressed in HEK293 cells[24], whereas parent OPN1MW and melanopsin are able to activate the Gαi-type G-proteins endogenously expressed in the HEK293-GIRK cell line. Similar effects could explain reduced GIRK currents elicited by Mela(CT+IL3mGluR6) ($44.0 \pm 3.7$ pA/pF, $n = 5$, mean ± SEM), which were significantly smaller compared to those elicited by Mela(CTmGluR6) ($98.0 \pm 9.7$ pA/pF, $n = 5$; $p = 0.002$). Notably, the mGluR6 C-terminus significantly increased GIRK-current densities in OPN1MW [OPN1MW(CTmGluR6): $78.0 \pm 9.6$ pA/pF, $n = 14$; OPN1MW: $46.1 \pm 7.8$ pA/pF, $n = 13$; $p = 0.027$] and in melanopsin [WT melanopsin: $22.0 \pm 5.7$ pA/pF, $n = 5$; $p = 0.0005$

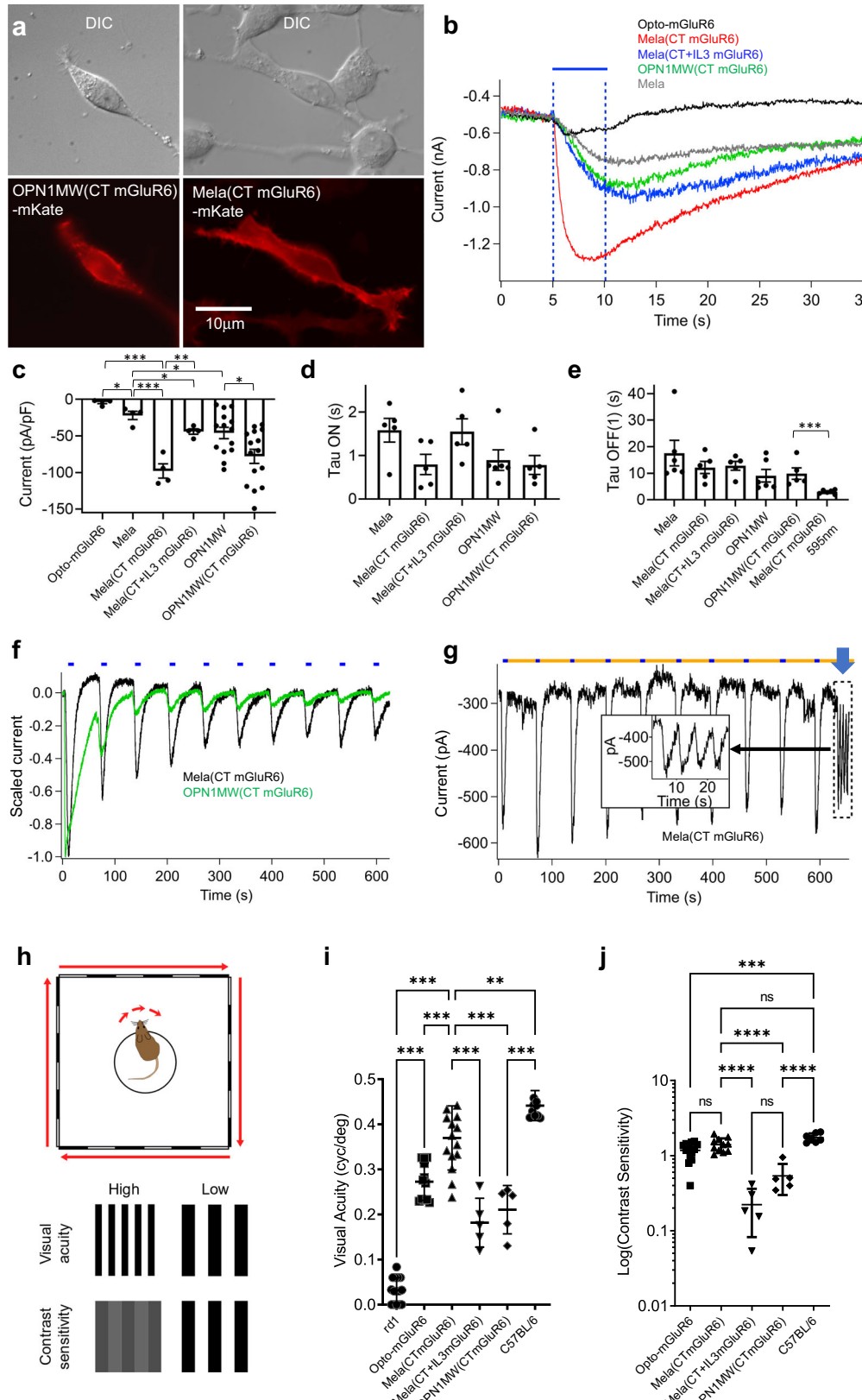

compared to Mela(CTmGluR6)], inferring that the mGluR6 C-terminus enhances expression of activatable opsin. All constructs elicited GIRK currents with similar ON time constants [Fig. 1d; Mela(CTmGluR6): 0.7 ± 0.2 s, n = 5; Mela(CT+IL3mGluR6): 1.6 ± 0.3 s, n = 5; OPN1MW(CTmGluR6): 0.8 ± 0.2 s, n = 6; Mela WT: 1.6 ± 0.3 s, n = 5] and OFF time constants

[Fig. 1e; Mela(CTmGluR6): 12.1 ± 2.3 s; n = 6; Mela(CT+IL3m-GluR6): 12.8 ± 1.7 s, n = 5; OPN1MW(CTmGluR6): 9.8 ± 2.2 s, n = 6; Mela WT: 17.6 ± 4.8 s, n = 6], providing evidence that melanopsin, without its regulatory C-terminus, is not inherently slow. Unlike native melanopsin, GIRK currents triggered by all chimeric constructs inactivated completely with a single

**Fig. 1 In vitro and in vivo screening of opsin constructs. a** HEK293-GIRK cells transiently transfected with different Opsin-mKate fusion proteins show robust membrane expression (exemplified for OPN1MW(CT mGluR6)-mKate and Mela(CTmGluR6)-mKate). Scale bars 10 μm. **b–h** To assess the relative efficacies at which opsin constructs activate G-proteins in the Gαi/o class, we screened all opsin constructs in HEK293-GIRK cells. **b** Example light responses recorded in HEK293-GIRK cells (stimulus presented by blue bar; 470 nm; $10^{14}$ photons/cm$^2$/s). **c** Average normalized GIRK response amplitudes recorded under the same conditions as in (**b**). Substituting the C-terminus with that of mGluR6 resulted in significantly larger GIRK responses to light in melanopsin ($p = 0.0008$) and OPN1MW ($p = 0.0159$) [$n$(Opto-mGluR6) = 4 cells, $n$(Mela) = 4, $n$(Mela(CTmGluR6)) = 4, $n$(Mela(CT +IL3mGluR6)) = 4, $n$(OPN1MW) = 14, $n$(OPN1MW(CTmGluR6)) = 15]. The relative Tau ON (**d**) and Tau OFF (**e**) values of the different opsins in HEK293-GIRK cells using similar data to that presented in **b**. Although not significant, we observed a slight tendency towards shorter Tau ON values in constructs that generated larger response amplitudes (**d**) [$n$(Mela) = 5, $n$(Mela(CTmGluR6)) = 5, $n$(Mela(CT+IL3mGluR6)) = 5, $n$(OPN1MW) = 6, $n$(OPN1MW(CTmGluR6)) = 5]. The primary time constants of the OFF response (Tau OFF(1)) did not differ significantly, with the exception of melanopsin chimeras recorded with an orange backlight (**e**; 595 nm at $5 \times 10^{15}$ photons/cm$^2$/s; $p = 0.0008$ [$n$(Mela) = 6, $n$(Mela(CTmGluR6)) = 5, $n$(Mela(CT +IL3mGluR6)) = 5, $n$(OPN1MW) = 6, $n$(OPN1MW(CTmGluR6)) = 5, $n$(Mela(CTmGluR6) 595 nm) = 6]. However, native melanopsin possesses a very slow second time constant (Tau OFF (2), see Supplementary Fig. 2). **f, g** Repetitive light stimuli (5 s duration at 1 min intervals; 470 nm; $5 \times 10^{13}$ photons/cm$^2$/s) in GIRK cells transfected with OPN1MW(CTmGluR6) (**f**; green) and Mela(CTmGluR6) (**f, b**; black). Since melanopsin is tri-stable, we observed no bleach run-down when using an orange backlight (**g**; 595 nm at $5 \times 10^{15}$ photons/cm$^2$/s). This response showed little attenuation, even at 0.2 Hz stimulation (blue arrow; insert). **h–j** In vivo screening of treated blind (>24 weeks) *rd1* mice in a naïve optomoter reflex task. The principle of the OMR task is depicted in panel (**h**). **i** Visual acuity thresholds: all opsins restore visual acuity significantly in blind *rd1* mice, with Mela(CTmGluR6) significantly outperforming the other constructs [$n$(rd1) = 15 animals (negative blind control), $n$(Opto-mGluR6) = 12, $n$(Mela(CTmGluR6)) = 16, $n$(Mela(CT +IL3mGluR6)) = 5, $n$(OPN1MWW(CTmGluR6)) = 5, $n$(C57BL/6) = 10 (positive seeing control)]. **j** Contrast Sensitivity thresholds: Mela(CTmGluR6) and Opto-mGluR6 significantly outperform the other constructs, with average contrast sensitivity values for Mela(CTmGluR6) treated mice not being significantly different to those determined in C57BL/6 mice. Since *rd1* mice were unable to track at 100% contrast, they are not depicted in this graph. [$n$(Opto-mGluR6) = 14, $n$(Mela(CTmGluR6)) = 11, $n$(Mela(CT+IL3mGluR6)) = 5, $n$(OPN1MW(CTmGluR6)) = 5, $n$(C57BL/6) = 7 (positive seeing control)]. Data in panels **c–e** depicted as means ± SEM, in **i** and **j** depicted as means ± SD.

exponential decay after termination of the light signal (Supplementary Fig. 2).

Next, we compared the relative bleach stabilities of Mela(CTmGluR6) and OPN1MW(CTmGluR6), an imperative property when considering a role in vision restoration. In particular, photoreceptor degeneration is often associated with pathologies of the retinal pigment epithelium resulting in a compromised visual cycle and a restricted supply of cis retinal, which may impact the function of monostable opsins such as OPN1MW. Melanopsin, a tri-stable pigment, is able to recycle its chromophore, a property shared by many invertebrate and microbial relatives (i.e., ChR2) that imparts bleach resistance. To demonstrate this difference between OPN1MW and melanopsin chimeras, we presented cells with consecutive light flashes (470 nm; $5 \times 10^{13}$ photons/cm$^2$/s) without providing 9-cis retinal. As expected, cells expressing OPN1MW(CTmGluR6) had a rapid response rundown compared to Mela(CTmGluR6) (Fig. 1f). When stimulated with an orange backlight (595 nm; $5 \times 10^{15}$ photons/cm$^2$/s), cells expressing Mela(CTmGluR6) retained full response amplitude, even at high light stimulation frequencies (0.2 Hz; Fig. 1g). The backlight also accelerated the OFF-kinetics of Mela(CTmGluR6) ($2.9 \pm 0.3$ s; $n = 6$; $p = 0.0008$, Fig. 1e and Supplementary Fig. 3) to values significantly faster than those of monostable OPN1MW(CTmGluR6) ($p = 0.007$), a potential further advantage of using a bi-stable optogenetic tool.

**Functional screening by the optomoter reflex (OMR).** We have previously established automated optomoter reflex (OMR) screening as a fast and reliable in vivo readout for optogenetically restored visual function[6,8]. The OMR is based on the vestibulo–ocular reflex that evokes head movements to stabilize an image on the retina and is driven mainly by ON-direction-selective ganglion cells (ON-DSGCs)[25]. We tested the performance of all four opsin constructs in restoring naïve behavioral OMR responses when expressed in the OBCs of the degenerated *rd1* mouse retina. Although the OBC population, particularly in the degenerated retina, was proven difficult to transfect[26], recent molecular advances, such as synthetic AAV capsids and cell-specific short promoters now enable efficient and selective OBC targeting in vivo[8]. We packaged all opsin construct in the AAV2.7m8 vector[27] under control of the 770En_454P(h*GRM6*)

promoter[8]. We injected blind *rd1* mice bilaterally and intravitreally with titer-matched vectors ($10^{10}$ vg/eye) and assessed restored visual function at >24 weeks of age when photoreceptors and light-responses have completely disappeared in untreated *rd1* littermates[28]. By determining the spatial frequency or contrast thresholds at which drifting sinusoidal gratings trigger head movements, we inferred both, the visual acuity and contrast sensitivity of the OMR[29] (Fig. 1h).

Figure 1i summarizes the visual acuity thresholds reached by the different treatment groups and relates them to average values achieved by sighted C57BL/6J mice ($0.44 \pm 0.063$ cyc/deg, $n = 10$) and untreated *rd1* littermates ($0.03 \pm 0.03$ cyc/deg, $n = 15$). All four opsin constructs significantly restored visual acuity in treated *rd1* mice. Mela(CTmGluR6) ($0.37 \pm 07$ cyc/deg, $n = 16$) significantly outperformed all other variants, including Mela(CT+IL3mGluR6) ($0.18 \pm 05$ cyc/deg, $n = 5$, $p < 0.001$), OPN1MW(CTmGluR6) ($0.21 \pm 0.05$ cyc/deg, $n = 5$, $p < 0.001$) and Opto-mGluR6 ($0.27 \pm 0.04$ cyc/deg, $n = 12$, $p < 0.001$).

We next determined contrast sensitivity thresholds by decreasing the luminance levels between the stripes at the most sensitive spatial frequency of our treated *rd1* mice (0.125 cyc/deg, Supplementary Fig. 4). Untreated *rd1* mice were unable to perform this task and were therefore excluded from the analysis and the graph shown in Fig. 1j. Mela(CTmGluR6) ($4.5 \pm 0.8\%$ Michelson contrast, $n = 11$, mean ± s.e.m.) significantly outperformed Mela(CT+IL3mGluR6) ($62.4 \pm 8.6\%$ Michelson contrast, $n = 5$, $p < 0.001$) and OPN1MW(CTmGluR6) ($32.0 \pm 5.8\%$ Michelson contrast, $n = 5$, $p = 0001$) treated *rd1* mice, but not Opto-mGluR6 treated mice ($9.0 \pm 3.5\%$ Michelson contrast, $n = 10$, $p = 0.246$). Remarkably, the contrast sensitivities restored by Mela(CTmGluR6) were not significantly different to those of normal sighted C57BL/6 mice ($2.1 \pm 0.4\%$ Michelson contrast, $n = 7$, $p = 0.128$). We observed a clear positive correlation between restored contrast sensitivities and restored visual acuities in opsin-treated *rd1* mice, similar to WT C57BL/6 mice, except in OPN1MW(CTmGluR6)-treated *rd1* mice (Supplementary Fig. 5).

Notably, the visual acuity and contrast sensitivity thresholds did not differ significantly in *rd1* mice that were tested at an older age (Supplementary Fig. 6) which is in line with previously reported long-term functional preservation of optogenetic gene therapies[30]. These OMR data did not directly reflect the result of

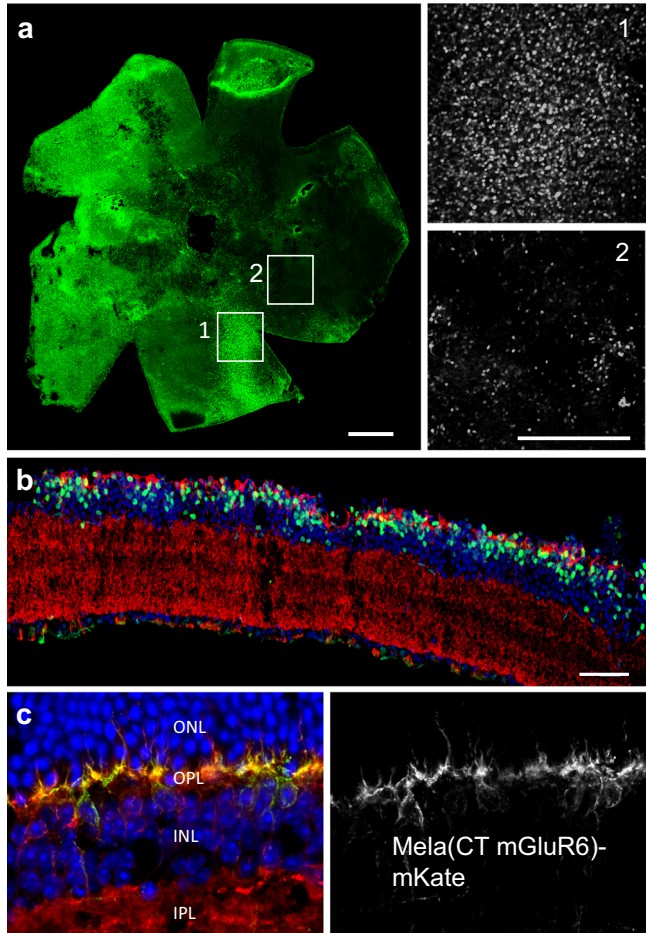

**Fig. 2 Targeted expression in OBCs after intravitreal injection.**
**a** A representative *rd1* retina four weeks after intravitreal injection of Mela(CTmGluR6)-IRES-TurboFP635. Inserts show higher magnification scanning micrographs taken from the bipolar cell layer at specified regions, with relatively high (1) or low (2) transduction. Despite the highlighted heterogeneity, OBCs were transduced across the retina. **b** Expression was specific for OBCs in the *rd1* retina (TurboFP635 in green; Gαo in red; DAPI in blue). **c** A Mela(CTmGluR6)-mKate fusion protein demonstrates robust targeted expression to the OBC dendrites (mKate in green; Gαo in red; DAPI in blue). Scale bars 500 µm in (**a**) and 50 µm in other panels.

the HEK-GIRK screening; Opto-mGluR6 outperformed Mela(CT +IL3mGluR6) ($p < 0.05$), whereas the performance of Mela(CT +IL3mGluR6) and OPN1MW(CTmGluR6) was not significantly different ($p = 0.999$). This may reflect differences in the in vivo scenario, such as modifications in protein expression or protein folding or differences in coupling to the Gαo (OBCs) and Gαi (HEK293 cells) G-protein subunit subtypes, as described above. Restored function also varied more in treated *rd1* mice compared to the controls, probably since the gene therapy typically leads to a heterogeneous opsin expression pattern (Fig. 2a) with varying efficacy in different treated animals.

**Mela(CTmGluR6) mediated light signaling in OBCs.** The behavioral results indicate that light responses in the inner retina are restored. For more detailed investigations on the quality of retinal signaling, we selected Mela(CTmGluR6) since it restored vision with the highest sensitivity in the OMR.

Retinal explants from treated *rd1* mice showed pan-retinal expression of the optogenetic protein, albeit patchy in nature,

with areas where almost all cells expressed and areas with much fewer expressing cells (Fig. 2a). Nevertheless, overall approximately half of OBCs in the *rd1* retina expressed Mela(CTmGluR6) ($55.4 \pm 11.6\%$) and expression was selective for OBCs ($64.9 \pm 3.9\%$, Fig. 2b). These results were encouraging for late degenerated tissue that has undergone neuronal rewiring, gliosis, and transcriptomic changes[31,32].

GPCR signaling is organized in subcellular domains with various intracellular interacting partners shaping the response[12,33]. To study the subcellular location of Mela(CTmGluR6), we expressed the Mela(CTmGluR6)-mKate fusion protein, which enabled direct observation of expression. In the wild-type retina, Mela(CTmGluR6)-mKate targeted to the dendrites and showed strong co-localization with Gαo, its primary interaction partner (Fig. 2c). We observed a similar expression pattern when expressing a mouse variant of Mela(CTmGluR6) that was visualized with an anti-mouse melanopsin antibody (Supplementary Fig. 7). In the degenerated *rd1* retina, Mela(CTmGluR6) trafficked predominantly to the cell bodies of the OBCs, as previously described for mGluR6 and the effector TRPM1 channel after the loss of photoreceptor input[34] (Supplementary Fig. 8). These immunohistochemical data confirm that Mela(CTmGluR6) is suitably located to couple into the endogenous mGluR6 signaling interactome in OBCs.

To functionally probe activation of the Gαo pathway within the OBCs, we performed cell-attached patch-clamp recordings from isolated OBCs and from OBCs in retinal whole mounts of *rd1* retinas. We targeted isolated OBCs from transduced C57BL/6 mice by their characteristic morphology under IR-DIC optics (Fig. 3a) and by fluorescent reporter expression (TurboFP635). The average membrane potential of isolated Mela(CTmGluR6)-expressing OBCs ($-27.8 \pm 1.8$ mV, $n = 18$) was not significantly different to that of non-transduced OBCs ($-28.8 \pm 2.7$ mV, $n = 38$; $p = 0.667$), indicative that Mela(CTmGluR6) did not change the properties of OBCs, as recently shown[35]. Only Mela(CTmGluR6)-expressing OBCs consistently responded with hyperpolarization to a full-field blue light stimulus (1 s duration, 470 nm, $5 \times 10^{14}$ photons/cm²/s). Most OBCs responded with a relatively sustained response (presumably rod-type OBCs, Fig. 3b), whereas one cell, presumably a cone-type OBC, responded more transiently with very fast ON-kinetics (TauON: 90.0 ms, TauOFF: 160.6 ms, Fig. 3c). One possible explanation would be that diverse intrinsic processing of the Mela(CTmGluR6) signal introduces a segregation of response types already at the OBC level.

To investigate the temporal properties of OBC responses in a more relevant setting, we next recorded from OBCs in the whole mount preparation of fully degenerated *rd1* retinas where synaptic circuits (feed-back and feed-forward) remain intact (Fig. 3d). The OFF response kinetics of rod-type OBCs was significantly accelerated compared to the isolated configuration (Fig. 3e, TauOFF wholemount: $715 \pm 350$ ms, $n = 9$; TauOFF isolated: $1726 \pm 985$ ms, $n = 11$, $p = 0.0001$), but no significant difference was found for the ON kinetics (TauON wholemount: $339 \pm 164$ ms; TauON isolated: $383 \pm 202$ m, $p = 0.303$). Acceleration of the OFF kinetics agrees with the action of inhibitory amacrine cell feedback onto the OBC terminals in the whole mount situation. Confirming melanopsin's relative bleach resistance, responses could be triggered repeatedly without response rundown at 0.4 Hz in the absence of cis-retinal supplementation (Fig. 3f).

To compare Mela(CTmGluR6) and photoreceptor-mediated responses in OBCs, we also recorded from dark-adapted retinal slices of C57BL/6 retinas. Light flashes reliably triggered depolarizations in C57BL/6 OBCs (Fig. 3g, red trace) as opposed to the hyperpolarizations observed in Mela(CTmGluR6)-expressing *rd1* OBCs (Fig. 3g, black trace). Inversion of the signal is a

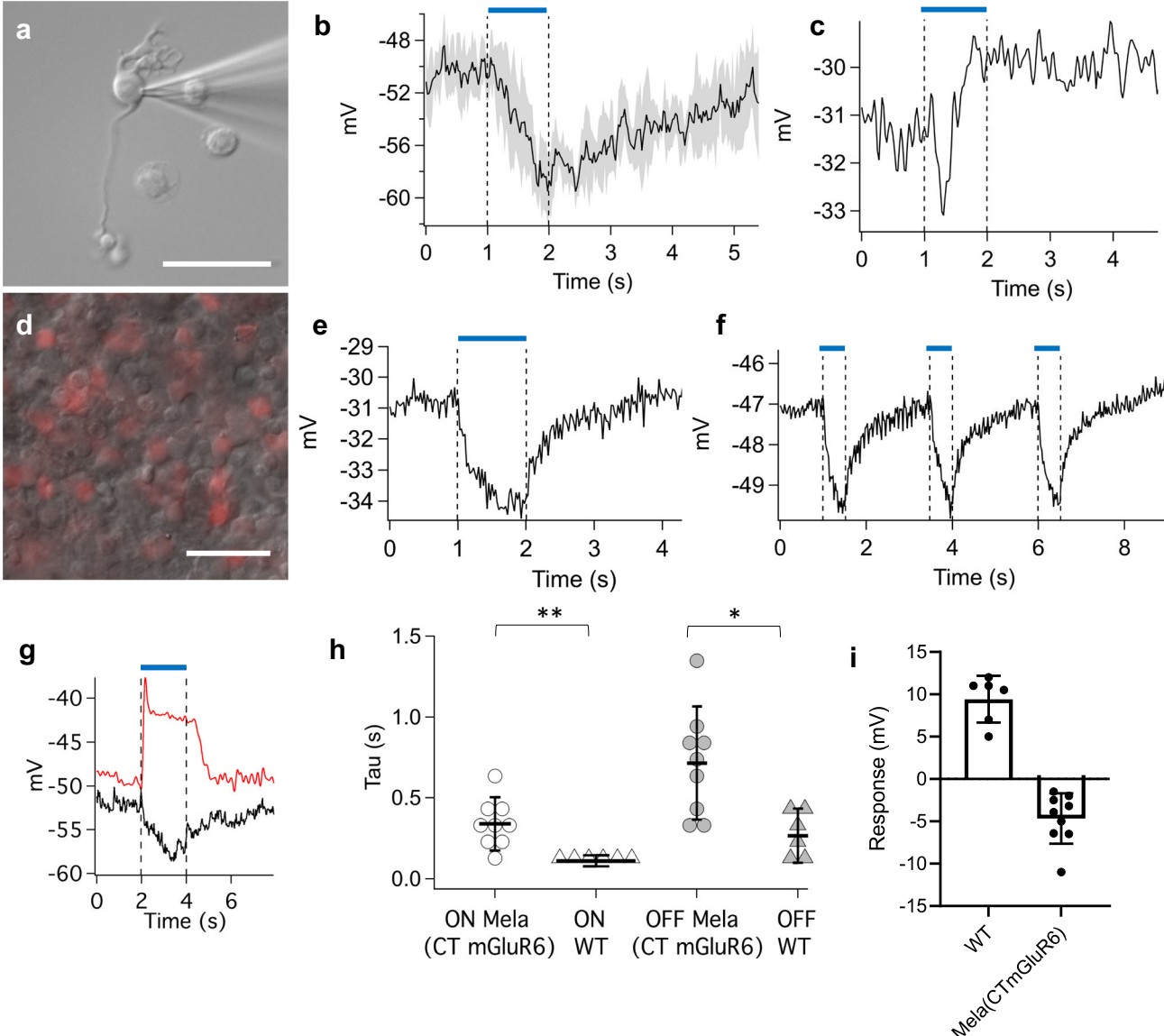

**Fig. 3 Mela(CTmGluR6) driven light responses in OBCs. a** We targeted isolated OBCs for electrophysiological recordings by their characteristic morphology under IR-DIC optics and reporter gene expression. **b** Isolated rod bipolar cells responded with a small but consistent hyperpolarization to light (470 nm; $10^{14}$ photons/cm$^2$/s; trace shows average response of 6 cells with shaded area indicating 1 SD). **c** An isolated cone OBC with a fast, biphasic response to the same light stimulus presented in B possibly suggests diverse intrinsic processing within different OBC types. **d** In whole mount *rd1* retinas, OBCs were targeted for recording by reporter gene expression and by the characteristic size and location of their cell bodies. Recordings from transfected rod bipolar cells in the whole mount *rd1* retina had robust light responses with a faster recovery time compared to isolated cells (**e**; 470 nm; $10^{13}$ photons/cm$^2$/s) and could be repeatedly triggered at 0.4 Hz (**f**). **g** Light responses from a transduced rod bipolar cell in a *rd1* whole mount retina (black) compared to a light response form a rod bipolar cells in a wild type C57BL/6 slice preparation of the retina (red; 470 nm; $1 \times 10^{13}$ and $1 \times 10^{11}$ photons/cm$^2$/s, respectively). **h** Tau ON and Tau OFF values of wild type OBCs ($n = 6$) and Mela(CTmGluR6)-transduced *rd1* OBCs ($n = 9$). Although time constants recorded from wildtype cells were on average faster, both, the ON and OFF time constants of some treated cells overlapped with those recorded from wildtype cells ($p = 0.0055$ and 0.0125, respectively). **i** Average response amplitudes of the same responses presented in (**g**) [$n$(WT) = 6, n(Mela(CTmGluR6) = 9]. Scale bars in **a** and **d** 50 μm. Data in panels **h**, **i** depicted as means ± SD, dark points represent individual cells.

clear indicator that Mela(CTmGluR6) acts through the mGluR6 signaling cascade. Glutamate is naturally released from the photoreceptors in the dark to activate mGluR6 while Mela(CTmGluR6) is inversely activated by light (Supplementary Fig. 9)[6]. While Mela(CTmGluR6)-driven OBC responses from the whole mount *rd1* retina were overall slower compared to photoreceptor-driven OBC responses recorded from dark-adapted C57BL/6 retinal slices ($n = 6$; wild-type TauON: $111 \pm 34.3$ ms, $p = 0.0055$; wild-type TauOFF: $266.8 \pm 167.4$ ms, $p = 0.0125$; Fig. 3h), there was some overlap, with several

Mela(CTmGluR6)-driven responses reaching the speed of wild-type responses. The larger scatter of kinetic values in Mela(CTm-GluR6) transduced *rd1* retinas likely reflects the variability in opsin expression within individual OBCs. To investigate how much "drive" Mela(CTmGluR6) can generate in OBCs of the *rd1* retina compared to the natural drive from photoreceptors, we compared the average response amplitudes in both scenarios. Light triggered changes in membrane potential were significantly larger in OBCs of the wild-type retina ($9.4 \pm 2.8$ mV, $n = 6$) compared to the *rd1* retina ($4.7 \pm 3$ mV, $n = 9$, $p = 0.0087$),

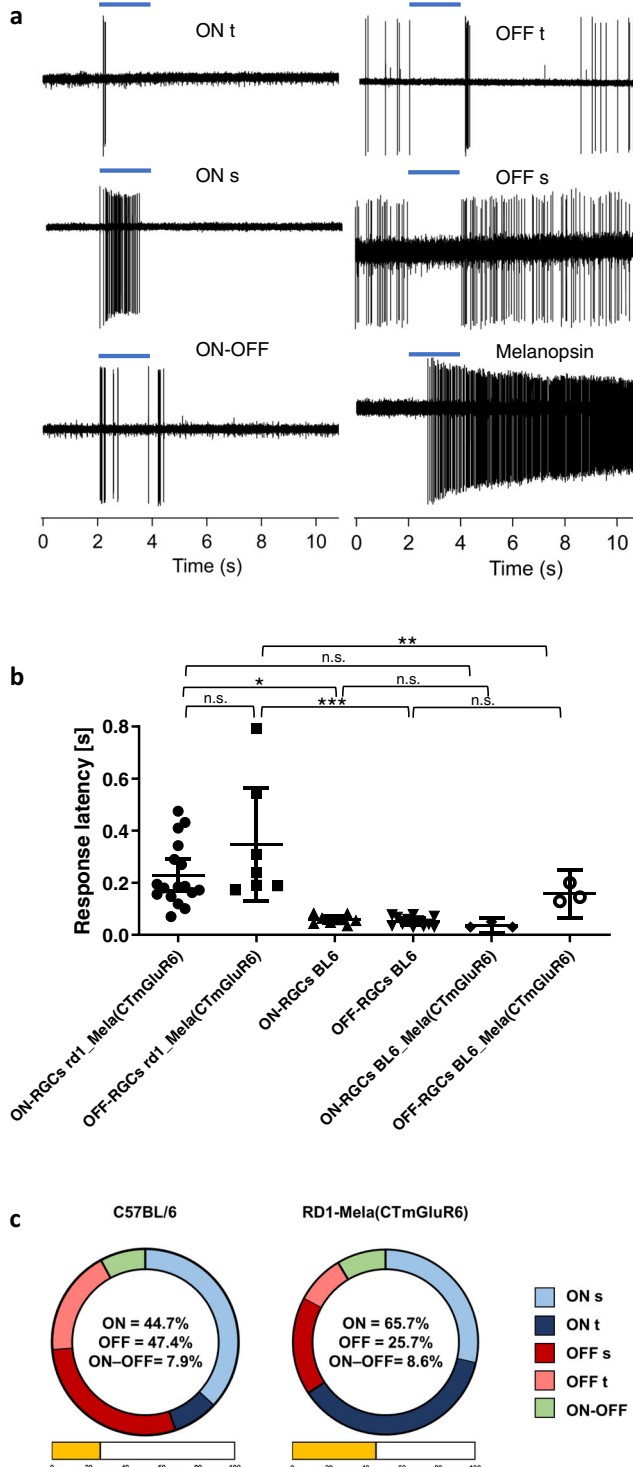

**Fig. 4 Mela(CTmGluR6) restores diverse receptive-field types in the RGC population of the degenerated *rd1* retina. a** Extracellular recordings from RGCs in the Mela(CTmGluR6)-treated *rd1* retina during light stimulation reveal restoring a diversity of RGC light-responses: transient ON (ON-T), transient OFF (OFF-T), sustained ON (ON-S), sustained OFF (OFF-S), ON–OFF as well as melanopsin responses from ipRGCs. **b** The response onset of ON and OFF RGCs in treated *rd1* retinas had a relatively wide scatter compared to the WT retina [n(ON-RGCs rd1_Mela(CTmGluR6)) = 17, n(OFF-RGCs rd1_Mela(CTmGluR6)) = 7, n(ON-RGCs BL6) = 8, n(OFF-RGCs BL6) = 12, n(ON-RGCs BL6_Mela(CTmGLuR6)) = 4, n(OFF-RGCs BL6_Mela(CTmGluR6)) = 3]. Data shown as means ± SD, individual data points depicted as black symbols. **c** Charts showing the relative fraction of receptive-field types (excluding melanopsin) recorded in the WT and the Mela(CTmGluR6)-treated *rd1* retina. Compared to the WT retina, we observed a relative shift from OFF to ON response types and from sustained to transient responses in the treated *rd1* retina.

patch-clamp recordings in whole mount retinas in combination with dye-injection to serve subsequent morphological identification of the RGC type and identified all cardinal RGC response types (Fig. 4a): ON-sustained (ON-S), ON-transient (ON-T), OFF-sustained (OFF-S), OFF-transient (OFF-T) and the ON-OFF type. To make sure that the response diversity was not attributed to off-target Mela(CT mGluR6) expression in RGCs, we specifically targeted RGCs with reporter (TurboFP635) expression and recorded light responses before and after pharmacological isolation (CNQX and D-AP5; 20 μM each) from the inner retina (Supplementary Fig. 10). Off-target expression was extremely low and we were able to record light responses from only three Turbo-FP635 expressing RGCs in 16 treated *rd1* retinas (from 8 animals). The light responses measured under synaptic block mirrored responses classified as ipRGCs (see below) and did therefore not contribute to the reported diversity of Mela(CT mGluR6) mediated RGC responses. We also labeled RGCs after recording with biocytin, which revealed that labeled RGC cells were typically bi-stratified confirming that light responses were not mediated by endogenously expressed melanopsin in ipRGCs. We therefore conclude that ectopic expression of Mela(CTmGluR6) in RGCs does not significantly contribute to the diversity of light responses reported to originate from Mela(CTmGluR6) expressed in OBCs. Finding ON-response types whilst eliciting hyperpolarizing responses in OBCs by Mela(CTmGluR6) confirms functional signaling through AII amacrine cells (Supplementary Fig. 9). Segregation of transient and sustained response types further infers a broadly restored inhibitory amacrine cell circuit within the inner retina. Remarkably, the diversity of RGC response types was preserved in retinas from very old *rd1* mice tested, indicative that the optogenetic treatment may stabilize or slow down the degenerative process (Supplementary Fig. 6).

We next compared the response latencies of ON and OFF type RGCs in the wild-type C57BL/6 and Mela(CTmGluR6)-treated *rd1* retinas. As for Mela(CTmGluR6)-driven responses in OBCs of the *rd1* retina, latencies were significantly attenuated and showed increased jitter (ON-RGCs: 0.23 ± 0.12 s, n = 17; OFF-RGCs: 0.35 ± 0.24 s, n = 7) compared to wild-type retinas (ON-RGCs: 0.06 ± 0.02 s, n = 8, p < 0.05; OFF-RGCs 0.05 ± 0.02 s, n = 12, p < 0.001; Fig. 4b). To investigate if response attenuation was due to degeneration-induced changes within the inner retina, such as cell death and rewiring[32], we also treated C57BL/6 mice with Mela(CTmGluR6). To isolate Mela(CTmGluR6)-mediated RGC responses in wild-type retinas, we blocked photoreceptor input pharmacologically with LAP-4 and by high intensity light bleaching $(5.35 \times 10^{17}$ photons/cm$^2$/s for 5 min)[6]. While we

but responses driven by Mela(CTmGluR6) notably reached about 50% of the wild-type amplitude (Fig. 3i). When considering the many pathological changes in the *rd1* retina, including loss of photoreceptors expressing a high abundance of photopigment in the disks of the outer segments, these data were encouraging.

**Detailed functional characterization of optogenetically restored retinal ganglion cell responses.** We next characterized the RGC full-field light-responses in the terminally degenerated *rd1* retina elicited by Mela(CTmGluR6). We used cell-attached

calculate a near total bleach of rhodopsin, it is important to note that an incomplete bleach may paradoxically preserve light sensitivity of rod photoreceptors by acting to prevent response saturation[36]. As evident from Fig. 4b, RGC response latencies of Mela(CTmGluR6)-mediated light responses in wild-type retinas were not significantly different to photoreceptor-mediated responses (ON-RGCs: $0.04 \pm 0.01$ s, $n = 4$; $p = 0.71$). These results infer that progressing retinal degeneration, including potential neuronal rewiring[6,31], affects signal propagation within the inner retina, and that the Mela(CTmGluR6) per se restores normal retinal signaling kinetics.

Comparing the overall distribution of RGC response types in treated *rd1* and wild-type retinas it became evident that ON-responses dominate in the treated *rd1* retina (68.7% ON and 25.7% OFF, $n = 37$), whilst ON and OFF responses show a more balanced distribution in the wild-type retina (ON: 44.7%, OFF: 47.4%, $n = 35$, Fig. 4c). Another difference observed in the treated *rd1* retina was a marked increase in transient responses (transient: 48.6%, sustained: 45.7%) compared to the wild-type retina where sustained responses dominated (transient: 26.6%, sustained: 65.8%). We interpret transientness as a consequence of strong activation of the inhibitory amacrine cell circuitry that truncates RGC responses.

Dye injection after recordings allowed subsequent anatomical identification of RGCs according to their arborization patterns in the distal and/or proximal stratum of the inner plexiform layer (Fig. 5). Interestingly, only 43% of the RGCs showed an inverse response polarity with respect to their anatomy and these cells were exclusively native OFF-RGCs (60% of OFF-RGCs, Supplementary Table 1). These results explain the accumulation of ON responses in the treated *rd1* retina compared to the wild-type retina. Since similar experiments in treated C57BL/6 mice previously showed a more consistent inversion of light response polarity[6], we interpret the partial lack of sign-inversion in the *rd1* retina as pathological changes within the amacrine cell network[37].

The single-cell patch-clamp data was supported by multi-electrode array (MEA) recordings from retinal flat-mounts of treated *rd1* mice. Significantly more RGCs exhibited light responses in Mela(CTmGluR6)-treated *rd1* retinas ($50.8 \pm 1$ $7.4\%$, $n = 24$; $p = 0.005$) compared to untreated *rd1* controls ($31.3 \pm 12.8\%$, $n = 11$retinas; Fig. 6a). These numbers however did not reach the percentage of light-responsive cells observed in healthy C57BL/6 retinas ($82.1 \pm 15.5\%$, $n = 12$, $p = <0.001$). Unsupervised spectral clustering, however, revealed six distinct RGC types in Mela(CTmGluR6)-treated *rd1* retinas known from wild type C57BL/6 retinas (Fig. 6b), with about 65% of responses starting during light stimulation (ON responses) and 35% after light stimulation (OFF responses; Supplementary Fig. 11a). These quantitative results support the more qualitative patch-clamp experiments and confirm the restoration of all cardinal retinal light response types in the optogenetically treated *rd1* retina, OFF and ON responses with different kinetic properties (transient vs. sustained). Kinetically slower ipRGC-like light responses were also found in Mela(CTmGluR6)-treated *rd1* retinas (Fig. 6b right, Supplementary Fig. 11b; Tau ON = $1.55 \pm 0.26$ s) and did not significantly differ from ipRGC responses observed in untreated *rd1* controls ($1.43 \pm 0.42$ s; $p = 0.078$). The same unsupervised response clustering was also performed on healthy controls and untreated *rd1* animals and yielded seven and two response clusters, respectively (Supplementary Fig. 11c, d). The peak firing frequencies in Mela(CTmGluR6)-treated *rd1* retinas were also similar to those in healthy C57BL/6 retinas ($5 \times 10^{14}$ photons/ cm$^2$/s; > 58 Hz; $p = 0.4058$) and significantly higher than those seen in untreated *rd1* controls ($p = 0.013$; Supplementary Fig. 11e). These observations may explain, at least in part, the high contrast sensitivities restored in the OMR recordings. We

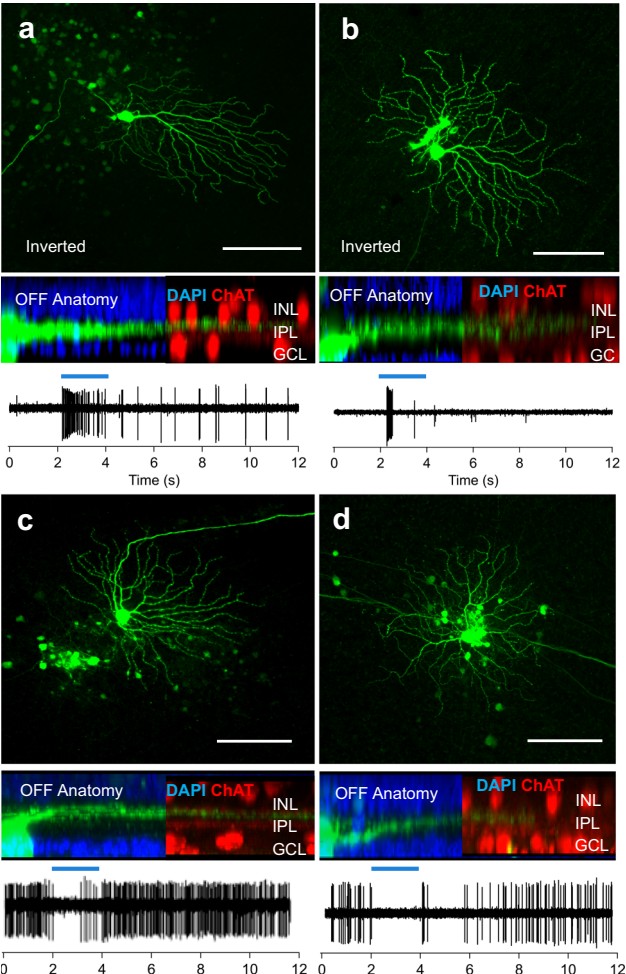

**Fig. 5 Anatomy-function correlation of Mela(CTmGluR6)-triggered RGC responses in the treated *rd1* retina determined by patch-clamp.** Anatomy and corresponding light responses of 4 example RGCs from Mela(CTmGluR6)-treated *rd1* retinas show that some native OFF RGCs (**a**, **b**) have inverted light responses (i.e., dendrites that stratify in the OFF sublamina of the inner plexiform layer (IPL) with a light response at the onset of light), while the light responses of other cells (**c**, **d**) are not inverted (i.e., cells that stratify in the OFF sublamina of the IPL respond at the end of the light stimulus). Also, see Supplementary Table 1. Light stimulus indicated as blue bar above spike trains (470 nm; 10$^{14}$ photons/ cm$^2$/s). Scale bars = 100 μm.

did not observe significant differences in response distributions between retinas from younger or older treated *rd1* animals (Supplementary Fig. 11f; $p = 0.703$).

Another fundamental property of vision is the ability to adapt retinal responsiveness to the range of ambient light intensities. We wondered whether light adaptation was still present in the optogenetically treated *rd1* retinas. For this, we generated dark- and light-adapted (15 min white light, 100 μW cm$^{-2}$) intensity-response curves from RGCs ($n = 62$ cells from 11 retinal explants) by exposing retinas to a series of brief (500 ms) 470 nm light flashes of increasing intensities. The dark-adapted intensity–response curve was in good agreement with previous data from single-cell RGC patch recordings[6] (Fig. 6c). The light-adapted curve showed a shift of approximately 1 log unit on the intensity axis, representing the light adaptive capability of the optogenetically restored inner retina. We also observed significantly higher spiking frequencies in the light adapted retina when using light intensities beyond the

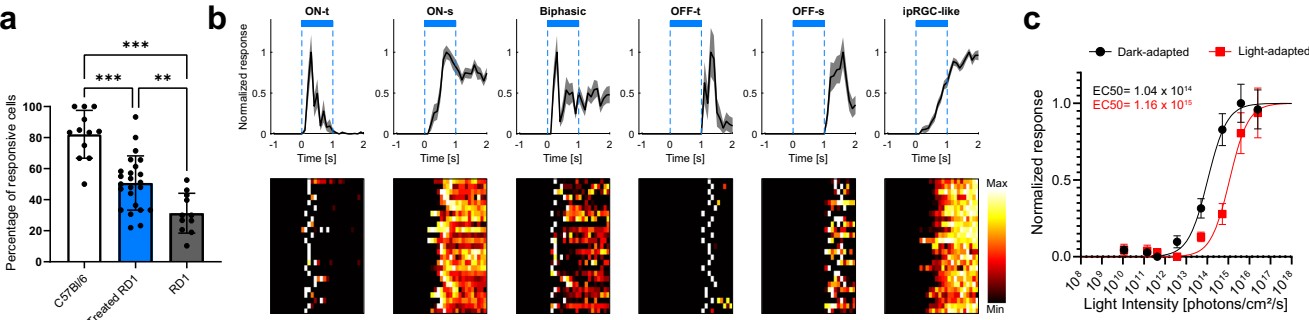

**Fig. 6 Compound RGC responses in retinas from Mela(CTmGluR6)-treated *rd1* mice determined on multi-electrode arrays. a** We observed a significantly higher percentage of responsive cells in treated *rd1* retinas ($n = 24$) compared to untreated *rd1* retinas ($n = 11$, $p = 0.005$). There was also significant difference in percentage of responsive cells between C57BL/6 retinas ($n = 12$) and Mela(CTmGluR6)-treated *rd1* retinas ($p < 0.001$). Data depicted as means ± SD, dark points represent individual values of retinal explants. **b** Example average traces of different functional RGC categories. Light stimulation (470 nm, $5 \times 10^{14}$ photons/cm²/s, 1 s) indicated as blue bars. Below each representative average trace (mean ± s.e.m.) is a heat map of representative individual cells. **c** Normalized dark-adapted (black trace) and light-adapted (red trace) intensity response curves from RGCs of Mela(CTmGluR6)-treated *rd1* retinas ($n = 62$ cells) fitted with a Hill curve (dark-adapted: slope = 0.9432, $EC_{50} = 1.04 \times 10^{14}$ photons/cm²/s; light-adapted: slope = 0.9794, $EC_{50} = 1.14 \times 10^{15}$ photons/cm²/s). Retinas were stimulated every 120 s with a blue light flash (470 nm, 500 ms, $1.12 \times 10^{10}$–$2.5 \times 10^{16}$ photons/cm²/s), light adapted for 15 min (white light at 100 µW cm⁻²), and again stimulated with identical blue light flash protocol. Data shown as means ± s.e.m.

half-saturation values (Supplementary Fig. 11g). Overall, the Mela(CTmGluR6)-expressing *rd1* retina responds over 6 log units of light intensities ($10^{10}$–$10^{16}$ photons/cm²/s), to our knowledge the widest range reported for optogenetic restoration of light sensitivity.

**Behavioral pattern vision and restoration of cortical light responses.** The OMR is a reflex elicited by a subset of direction-selective ganglion cells (DSGCs) that project to the accessory optic system—and not *via* the lateral geniculate nucleus (LGN) to the primary visual cortex (V1). We, therefore also probed cortical image forming vision behaviorally. We employed a conditioned visual task in which the mouse had to distinguish drifting gratings of the highest acuity achieved in the OMR, 0.35 cyc/deg (100% contrast, rotation speed 12°/s) from equiluminescent gray. The mice were conditioned in a custom-made plexiglass shuttle-box consisting of two identical adjoining compartments connected by a small opening (Fig. 7a). Each compartment was equipped with a computer monitor that displayed either gray with a luminance equivalent to the OMR ($5 \times 10^{13}$ photons/cm²/s) or an equiluminescent drifting grating (0.35 cyc/deg). On the first day, mice were habituated to the shuttle-box (Fig. 7a). During a 2-day training period, the screen switched three times every 5 min from gray to the equiluminescent pattern stimulus in the compartment where the mouse resided at the time. The stimulus was paired with an aversive foot shock in that same compartment. On day 4 the electric grids were removed and the odor, as well as the orientation of the shuttle-box, changed to avoid environment-induced conditioned fear behavior. The drifting grating was displayed on the side where the mouse resided after a 5 min habituation period. Aversive behavior during the display of the pattern stimulus was compared to the baseline aversive behavior before stimulus onset to get a measure for the behavioral change as a consequence of pattern recognition. The increase of aversive behavior in response to the visual cue was highly significant in Mela(CTmGluR6)-treated ($p = 0.0003$) and C57BL/6 mice ($p = 0.0026$), but no behavioral change in response to the drifting stimulus was seen in untreated *rd1* littermates ($p = 0.83$, Fig. 7b). Mela(CTmGluR6)-injected *rd1* mice ($n = 9$) displayed strong visually-cued aversive behaviors similar in magnitude to the wild-type C57BL/6 mice (Fig. 7c, $n = 6$, $p = 0.9935$, ANOVA with post-hoc Tukey).

From the same mice, we subsequently recorded visually evoked field potentials (VEPs) from layer 4 of V1. A 500 ms blue light stimulus of similar light intensity employed in the OMR and open field box paradigm ($5 \times 10^{13}$ photons/cm²/s) was supplied to the contralateral eye. We compared responses from Mela(CTm-GluR6)-treated and non-treated *rd1* mice older than 280 days of age when V1 was shown to no longer respond to visual stimuli in *rd1* mice[1]. In addition, we recorded VEPs from C57BL/6 dark- and light-adapted mice. VEPs were reliably evoked in C57BL/6 (both dark- and light-adapted) and Mela(CTmGluR6)-treated *rd1* mice, but not in non-treated *rd1* littermates (Fig. 7d). VEPs from Mela(CTmGluR6)-treated *rd1* mice were similar in amplitude ($505.5 \pm 161.9$ µV) to the VEPs of light-adapted C57BL/6 retinas ($366.9 \pm 89.8$ µV; $p = 0.043$; Fig. 7e). This aligns with previous studies showing that light-adapted VEPs in the mouse have a smaller amplitude compared to dark-adapted VEPs[38]. The response latencies of Mela(CTmGluR6)-treated *rd1* mice ($555.6 \pm 26.6$ ms), however, were significantly increased compared to the response latencies of both, dark-adapted ($132.4 \pm 17.7$ ms, $p < 0.0001$) and light-adapted C57BL/6 mice ($107.8 \pm 28.8$ ms, $p < 0.0001$; Fig. 7f). To investigate if this delay was due to retino-cortical signal processing or originated in the degenerating retina, we performed MEA recordings on Mela(CTmGluR6)-treated *rd1* retinas with matched light intensities ($5 \times 10^{13}$ photons/cm²/s, see Fig. 6c and Supplementary Fig. 11g). Under these conditions we elicited robust, but delayed RGC light responses with latencies similar to the VEPs measured under the same conditions ($481 \pm 254$ ms; $n = 41$, $p = 0.434$; Fig. 7f). This provides evidence that the delay of VEPs in V1 of treated *rd1* mice is an effect of retinal degeneration and not due to altered retino-cortical signaling in the *rd1* mouse.

## Discussion

Optogenetic gene therapies are in clinical trials to restore vision in blind patients (ClinicalTrials.gov Identifier: NCT03326336, NCT02556736, NCT04278131). Recently, a milestone was reached when one patient with advanced retinitis pigmentosa that was treated with a therapy introducing the microbial channelrhodopsin, ChrimsonR, into the RGCs has regained the ability to locate and count objects[16]. Due to the very high light intensities required for ChrimsonR activation, the patient was additionally equipped with biomimetic goggles amplifying and spectrally-

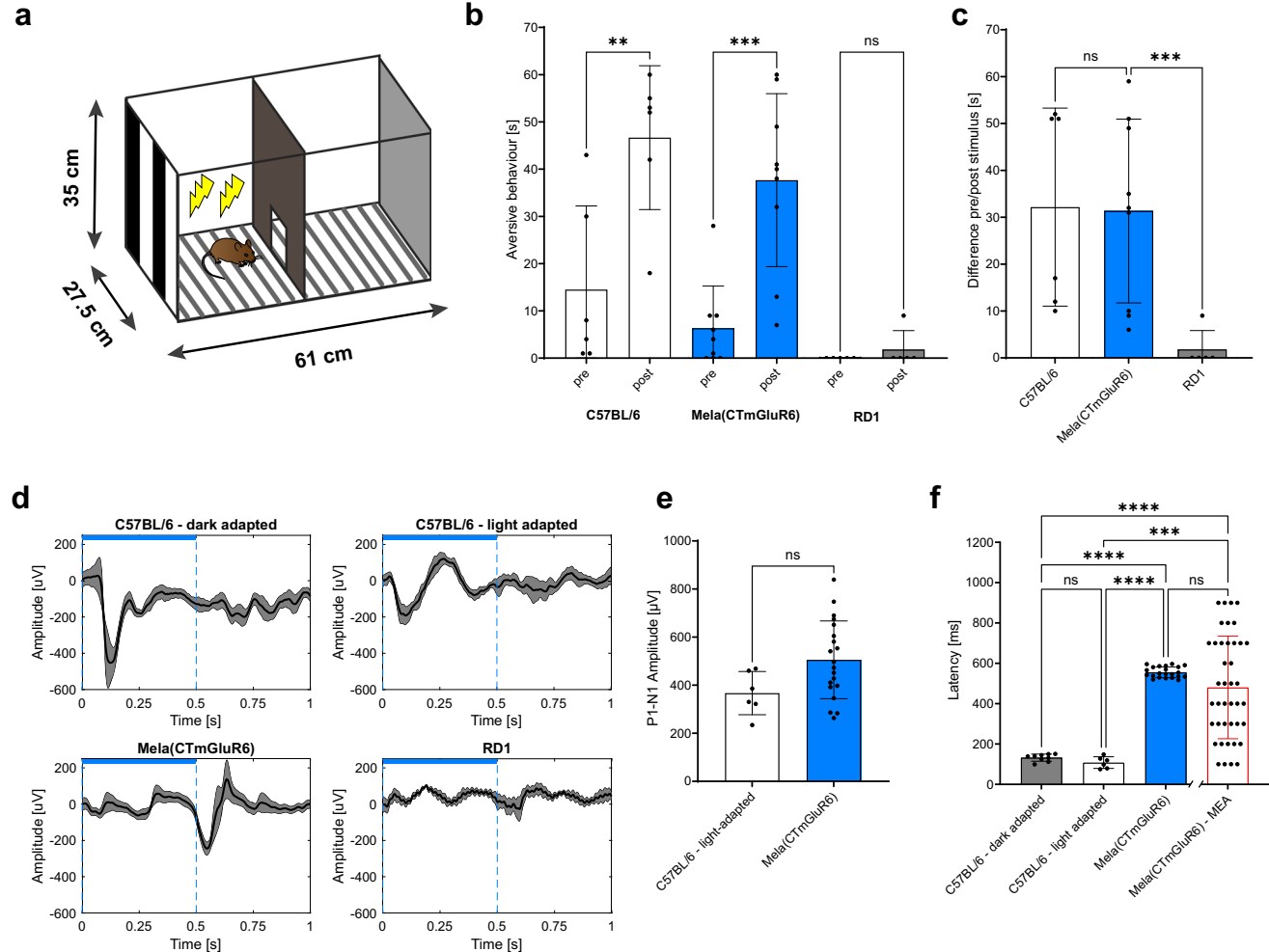

**Fig. 7 Restoration of conditioned visually guided behavior. a** Graphical depiction of the two-compartment visual conditioning paradigm to a moving grating (0.35 cyc/deg spatial frequency). **b** Comparison of aversive behaviors on day 4 before and during the pattern display increased significantly in C57BL/6 controls ($n = 6$ animals; $p = 0.00263$) and Mela(CTmGluR6)-treated *rd1* mice ($n = 9$ animals; $p = 0.00026$), while non-treated *rd1* mice did not show a behavioral response to the pattern stimulus ($n = 5$ animals; $p = 0.83$). See Supplementary Movie M1 in the supplementary information for a typical behavior of a treated *rd1* mouse in response to the drifting gratings. Comparison of 60 s pre-stimulus vs. 60 s during stimulus. **c** Stimulus-triggered aversive behavior was equally observed in Mela(CTmGluR6) injected *rd1* mice ($n = 9$) and healthy C57BL/6 mice (positive control, $n = 6$; $p = 0.9935$), but not in untreated *rd1* littermates (negative control, $n = 5$; $p = 0.00005$). **d** Average VEP responses to a full-field light stimulus (470 nm, $5 \times 10^{13}$ photons/cm$^2$/s, 500 ms) in dark-adapted C57BL/6 animals ($n = 9$ traces), light-adapted C57BL/6 animals ($n = 6$ traces), Mela(CTmGluR6)-treated *rd1* animals ($n = 22$ traces) and untreated *rd1* controls ($n = 9$ traces). **e** Comparison of P1-N1 amplitudes in light adapted C57BL/6 animals ($367 \pm 90$ µV; $n = 6$ traces) and Mela(CTmGluR6)-treated *rd1* animals ($506 \pm 162$ µV; $n = 20$ traces). No significant differences were observed between experimental groups ($p = 0.123$; Kolomogorov–Smirnov test). **f** Comparison of response latencies. The latency was significantly increased in Mela(CTmGluR6) treated *rd1* animals ($556 \pm 27$ ms, $n = 22$ traces) in comparison to dark-adapted C57BL/6 mice ($132 \pm 18$ ms, $n = 9$ traces, $p < 0.000001$; one-way ANOVA) and light-adapted C57BL/6 mice ($108 \pm 29$ ms, $n = 6$ traces, $p = <0.000002$). No significant difference was observed between dark- and light-adapted C57BL/6 animals ($p = 0.99$). Latencies of light-intensity matched MEA recordings from RGCs of Mela(CTmGluR6) treated *rd1* mice ($481 \pm 254$ ms; $n = 41$ cells; 500 ms; 470 nm; $5 \times 10^{13}$ photons/cm$^2$/s) did not differ significantly from the V1 latencies in Mela(CtmGluR6) treated mice ($p = 0.434$). Data in bar plots shown as mean ± SD. Dots in plots represent individual measurements.

adapting the ambient light. Multiple approaches are currently under development with the goal to further improve the quality of optogenetically restored vision. Since neural processing within the retina extracts approximately 30 parallel information channels from the visual scene[7], endeavors are made to restore inner retinal processing by direct optogenetic targeting of inner retinal cells. Retinal processing is mostly lost when rendering RGCs light-sensitive using an optogenetic actuator. Therefore, OBCs, the first order retinal interneurons, are undoubtedly attractive targets for optogenetic vision restoration, since they feed the light signal into the inner retinal circuitry and thereby maximize the receptive-field diversity within the RGC population. OBC-targeted optogenetics

was recently made possible by molecular advances that allow efficient and selective OBC targeting[8], despite OBCs being rather non-permissive to AAV transduction[39].

In this study we investigated the potential of OBC-targeted optogenetic vision restoration in detail. We confirm that activation of OBCs restores inner retinal signaling and with that, RGC receptive-field diversity, an important building block for high quality vision. We focused on chimeric Opto-GPCRs that efficiently activate the native mGluR6 signaling cascade[6,40], making them 3–4 log units more light sensitive compared to their microbial counterparts (i.e., ChRs). We compared four Opto-GPCRs: Opto-mGluR6[6], OPN1MW-mGluR6[5,8] and two new

melanopsin/mGluR6 chimera variants with only the C-terminus (Mela(CTmGluR6)), or the C-terminus and IL3 (Mela(CT+IL3mGluR6)) replaced by that of mGluR6. Whilst C-terminal replacement facilitated subcellular trafficking and efficient expression, exchanging the IL3 domain appeared to compromise opsin function and/or expression. Although additional replacement of IL2 in melanopsin, as in Opto-mGluR6, further reduced the currents elicited in HEK293 cells (see Fig. 1b, c), this almost complete chimera design appeared to optimize coupling to Gαo, as evident from the well restored OMR (see Fig. 1i, j). As shown previously, additional replacement of IL1 did not improve function or Gαo coupling in Opto-mGluR6[6]. Opto-mGluR6 and Mela(CTmGluR6) both restored high contrast and spatial vision in treated *rd1* mice, with Mela(CTmGluR6) outperforming Opto-mGluR6 in the restoration of visual acuity, with some mice reaching performances of C57BL/6 mice. One possible explanation for the diverging results for Opto-mGluR6 in the HEK293-GIRK patch-clamp experiments and the in vivo OMR experiments is that HEK293-GIRK cells do not express the Gαo type G-protein and may therefore not accurately predict coupling to the mGluR6 pathway targeted in OBCs in vivo[24,41]. Opto-mGluR6 with the most complete mGluR6 intracellular domain exchange may couple very well into the Gαo pathway of OBCs, but less successfully into the Gαi pathway present in HEK293 cells. In this respect, the high complexity of GPCR signaling and regulation, which depends on the intracellular complement of cell-specific binding partners, may advocate the use of chimeric Opto-GPCRs over non-engineered opsins[14]. On the other hand, extensive chimera design may compromise opsin function. Our data show that the minimally chimeric Mela(CTmGluR6) has good Gαo coupling, expression and function. It is evident that Mela(CTmGluR6) efficiently activates the Gαo pathway within retinal OBCs, since: (1) The polarity of the Mela(CTmGluR6) signal in OBCs is inverted (hyperpolarizing) compared to the photoreceptor-driven light signal in the C57BL/6 retina (depolarizing), a consequence of direct Gαo light activation[33,42] (see Fig. 3g), (2) the restored light-responses are kinetically similar to those recorded in healthy C57BL/6 retinas (see Fig. 3h), (3) Mela(CTmGluR6) elicited cell-subtype specific light-responses in isolated rod and one cone OBCs, i.e., sustained vs. transient responses, indicative of activation of different endogenous signaling pathways as a result of differences in the molecular mGluR6 cascade elements in different OBC types (see Fig. 3c, d). Although temporal filtering continues in the inner retina, this early segregation of information into separate temporal channels supports the diversity of RGC output and is lost when expressing microbial opsins (e.g., ChR2) in the OBCs. At the RGC level, Mela(CTmGluR6) restored all cardinal RGC response types (ON-S, ON-T, OFF-S, OFF-T, ON–OFF) in response to a full-field light stimulus, indicative of sufficient optogenetic drive at the OBC level to efficiently activate the inner retinal circuitry.

We confirmed that Mela(CTmGluR6) trafficked to the macromolecular mGluR6 cascade complex in the OBC dendrites and co-localizes with Gαo and the effector channel TRPM1, fundamental for successful restoration of Gαo signaling in OBCs. The kinetics of optogenetically-elicited light-responses in OBCs were near to that of endogenous mGluR6. We speculate that this was achieved through Mela(CTmGluR6)'s ability to drive specific mGluR6 activity regulators, such as RGS proteins, $Ca^{2+}$ binding proteins, GRKs, and arrestins resident within OBCs.

The native light intensity integrator and circadian master clock entraining function of melanopsin is relatively slow, which so far reduced melanopsin's attractiveness for optogenetic control of precise neuronal activity[43,44]. However, we here show that truncated and C-terminally adapted melanopsin is an excellent and fast optogenetic tool as evident from the fact that Mela(CTmGluR6) supported vision at high spatial acuity in the OMR and open field box.

We based chimeras on melanopsin because cone opsin bleaches and depends on the visual cycle within the retinal pigment epithelium, which is often compromised in retinal degeneration. Melanopsin, similar to microbial opsins and some non-ciliary G-protein coupled opsins is bleach resistant, with a closed-loop photocycle that re-isomerizes the dark-state chromophore[45,46]. Consequently, and as shown, the activity of melanopsin does not decrease upon repeated stimulation in the presence of long-wavelength light (595 nm) and retains high amplitude also without the use of long-wavelength backlight. Temporally controllable light-modulation of the ON and OFF states makes melanopsin—and non-bleachable opsins in general—particularly attractive tools for optogenetic applications.

While the spike latencies and spike firing rates in RGCs produced by Mela(CTmGluR6)-input to OBCs of non-degenerated mice were similar to those elicited by photoreceptor input, latencies of light responses were strongly attenuated in the degenerated *rd1* retina, not only in the RGCs but also in V1. Our results point towards functional adaptations within the inner retina during the degenerative process. Indeed, previous anatomical studies showed that bipolar cells undergo remodeling in later stages of degeneration, including changes in morphology, synaptic connectivity, and receptor expression[32,34,47,48]. However and notably, our data hints towards a potential stabilization of the degenerative process in the *rd1* retina upon optogenetic treatment, since the restorative outcome did not deteriorate with time after treatment (see Supplementary Figs. 8 and 9f). This may indicate a protective effect of optogenetic activation; however, proving this would require further experiments.

We observed an accumulation of ON-type and transient RGC responses in the treated *rd1* retina. This was surprising since optogenetic activation of OBCs with Mela(CTmGluR6) leads to a hyperpolarization of OBCs corresponding to OFF responses in native ON-cells. The shift to more ON-type responses reflects our finding that exclusively OFF-RGCs seem to undergo sign-inversion, as demonstrated by the dendritic stratification patterns of recorded cells. Since ON-RGC responses are indirectly generated via AII amacrine cells (Supplementary Fig. 9), our findings confirm successful recruitment of the AII circuit. We interpret transientness to be a consequence of strong activation of the inhibitory amacrine cell circuitry within the inner retina that acts to truncate the RGC response[4]. Interestingly, an increase in transientness of RGC output was previously reported in a study where ChR2 was expressed in OBCs[4]. This suggests that strong inhibitory feedback may be a feature of the degenerated retina. The Mela(CTmGluR6) treated *rd1* retina also regained some ability to adapt to ambient light levels, which we also attribute to inner retinal circuits (Fig. 6c). Adding to these findings, the restoration of the OMR in treated *rd1* mice may infer more complex direction-selective circuits, although not proven here.

In summary, we find the restorative outcomes achieved by OBC-targeted optogenetic gene therapy encouraging, particularly since the encoding capacity of the inner retina appears to be largely preserved after optogenetic treatment, albeit with small variations that could arise from aberrant responses in the *rd1* retina most likely originating from lateral spread of activity[49] or changes in basal activity[50] Since our findings are particularly interesting from a translational point of view, alluding to improved strategies for optogenetic intervention, further physiological investigations of retinal circuit adaptations during the degenerative process in mice, and ideally in the human retina, are needed to fully interpret our results. Nevertheless, extrapolation of results from mice to human patients should be done with caution, and off-target expression rates have to be kept low in a bipolar cell targeted approach, since RGCs obviously possess Gi-

mediated intracellular pathways that can be hijacked by Mel-a(CTmGluR6) and other Gi-coupled optogenetic proteins

## Methods

**DNA and viral constructs**. All opsins were cloned into a pIRES2_opsin_TurboFP635 plasmid[6]. To create fusion proteins, the fluorescent protein, mKate2, was cloned into the position of IRES-TurboFP635. Opsin chimeras were generated using overlap extension PCR described in detail elsewhere[6]. Only the overhang primers for exchanging the C-terminus in pIRES_Mela(CTmGluR6) are given here, for other chimeric variants the overlap primers were adapted accordingly: CACCTGCCCTGCCTGTTC CATCCAGAGCAGAATGTGC (R1) and ATCATTTACGCCATCACCCACCCCAA GTACAGGGTGGCC (F2). The Kir2.1 membrane trafficking signal (TS) as well as the rhodopsin trafficking sequence (1D4 epitope) were added to all parent opsins and opsin-mGluR6 chimeras.

Viral vectors were produced in HEK293 cells by the triple plasmid co-transfection method using the pXX80 helper plasmid and the rep-cap plasmid encoding AAV2(7m8)[27]. Titers were all between $1 \times 10^{12}$ and $5 \times 10^{13}$ genome copies per ml.

**HEK293-GIRK whole-cell patch-clamp experiments**. A stable HEK293-GIRK1/2 cell line (kind gift from Olivia Masseck) was transiently transfected with the opsin constructs. After 24 h, whole-cell patch-clamp experiments were performed in a high potassium extracellular solution containing 60 mM KCl, 89 mM NaCl, 1 mM MgCl$_2$, 2 mM CaCl$_2$ and 10 mM HEPES, pH 7.4. Patch pipettes had a resistance of about 6 MΩ and were filled with an intracellular solution containing 140 mM KCl, 10 mM HEPES, 3 mM Na$_2$ATP, 0.2 mM Na$_2$GTP, 5 mM EGTA and 3 mM MgCl$_2$ (pH 7.4). Cells were voltage clamped at −70 mV while recording GIRK responses to various light stimuli using a HEKA EPC10 amplifier with PatchMaster software. To identify current responses by whole-cell patch-clamp in a high potassium extracellular solution triggered by a blue light stimulus (470 nm) generated by a pE-4000 system (CoolLED, Andover, United Kingdom), which was kept constant in length and intensity (5 s; $10^{14}$ photons/cm$^2$/s), and projected through a 20× water immersion objective onto the recorded cell. Traces were analyzed offline using Igor Pro software (version 7, Wave Metrics). TauON and TauOFF values were obtained by a single exponential fit. Current amplitudes were normalized to cell size (pA/pF).

**Animals experiments**. The experiments were performed on C3H/HeOuJ mice with retinal degeneration (rd1) and on C57BL/6 wild-type mice. Animal experiments and procedures were in accordance with the Swiss Federal Animal Protection Act and approved by the animal research committee of Bern (approval number BE99/19). Animals were maintained under a standard 12 h light-dark cycle.

Mice were bilaterally and intravitreally injected at 6–8 weeks of age (2.5 µl, $1 \times 10^{12}$–$5 \times 10^{13}$ vg/ml). For this, they were anesthetized by intraperitoneal injection of 100 mg/kg ketamine and 10 mg/kg xylazine. The pupil of the right eye was dilated with a drop of 10 mg/ml atropine sulfate (Théa Pharma). We then punctured the dorsal sclera approximately 1 mm from the corneal limbus using an insulin needle. The insulin needle was removed and a 33 G blunt needle was maneuvered through the pre-made hole to the back of the eye (RPE injection kit from World Precision Instruments). We then injected 2.5 µl of the rAAV vector solution and waited for 2 min before retracting the injection needle form the eye. The second eye was subsequently injected using the same procedure. Following surgery, an antibiotic eye lotion (Isathal from Dechra Veterinary Products) was applied to the eyes to prevent infection and drying of the cornea. Injected mice were kept under enhanced lighting provided by a Philips HF3319 daylight lamp (10,000lux) positioned 50 cm from the cage to guarantee sufficient light levels for activation of ectopic opsins.

For rd1 mice, injected and control littermates were physiologically tested at ≥ p168 (specific ages given in Supplementary Fig. 6)[6,39], except for recordings from V1, which were conducted at ≥p280. For patch-clamp and multi-electrode array recordings, mice were dark adapted for 60 min and subsequently sacrificed using isoflurane and cervical dislocation. Following enucleation, eyes were dissected under dim red light conditions in Ames medium (Sigma-Aldrich) that had been oxygenated for at least 60 min prior the procedure with carbogen (95% O2/ 5% CO2). After cortical recordings, animals were also sacrificed using isoflurane and cervical dislocation.

**Retinal patch-clamp recordings**. Bipolar cells were patch-clamped with electrodes of 8–10 MΩ using the perforated, cell-attached method either in whole mount retinas (rd1), in slices, or after acute dissociation. Bipolar cells were patch-clamped using the perforated, cell-attached method. Cells were patched either in whole mount retinas (rd1), in slices, or after acute dissociation. Slices were made by imbedding the retina in 1% low-melting agar in HEPES buffered Ames medium at 40 °C before immediately cooling the agar on an ice block. The solidified agar was then sectioned at a thickness of 250 µm on a Camden instruments vibratome. Isolated cells were prepared by incubating the retina for 45 min at 37 °C in Earle's Balanced Salt Solution supplemented with 40 units/ml papain (lyophilized, Worthington), 5 mM L-cysteine and 0.02% BSA. Papain digestion was followed by gentle titration with a glass pipette before plating cells on Poly-L-Ornithine coated coverslips. All preparations were patched-clamped in a recording chamber perfused with Ames medium (Sigma-Aldrich) at 34–36 °C. Patch electrodes were

pulled from borosilicate glass to a final resistance of 8–10 MΩ. The intracellular solution contained (in mM): KCL 110, NaCl 10, MgCl$_2$ 1, EGTA 5, CaCl$_2$ 0.5, HEPES 10, GTP 1, cGMP 0.1, ATP 1, and cAMP 0.05. Directly before the experiment, a saturated solution of Amphotericin B in DMSO was added to the intracellular solution at a 1:200 dilution. After adding the Amphotericin B, the solution was vortexed for 1 min and filtered before use. Transfected bipolar cells were identified using a fluorescent reporter (TurboFP635) and targeted for recording under vision control using IR-DIC optics. Light stimuli were generated similar to that described for the HEK-GIRK recordings above. Current recordings were made using a HEKA EPC10 amplifier with PatchMaster software. Traces were analyzed offline using Igor Pro software (Wave Metrics). TauON and TauOFF values were obtained by a single exponential fit.

The methods for recoding cell-attached light responses from RGCs have been described in detail previously[6]. In brief: Electrodes were pulled from borosilicate glass to a final resistance of 5–8 MΩ and filled with Ames medium. RGCs were targeted and approached under visual control using IR-DIC optics until spontaneous action potentials were observed in the voltage recording. Light stimuli were generated similar to that described for the HEK-GIRK recordings above. Voltage recordings were made using a HEKA EPC10 amplifier with PatchMaster software. To label RGCs, we patched cells in the whole-cell configuration using the same intracellular solution described for the HEK-GIRK cells above but supplemented with 0.2% biocytin (Sigma). The retina was subsequently fixed in 4% paraformaldehyde in 0.1 M phosphate buffer (pH 7.4) for 20 min. Alexa 488 conjugated to streptavidin was used to visualize biocytin-labeled cells (1:400; Invitrogen; S-11223).

All recordings were made using a HEKA EPC10 amplifier with PatchMaster software and performed at 34–36 °C.

**Immunohistochemistry**. At the end of the terminal experiment, mice were euthanized and retinas extracted for subsequent immunohistochemistry to confirm retinal expression patterns of the optogenes. Immunohistochemistry of cryosections were similar to that described previously[26]. In brief, retinas or eyecups were fixed in 4% paraformaldehyde in 0.1 M phosphate buffer (pH 7.4) for 30 min. Antibodies were diluted in a blocking solution containing 1% Triton-X and 2% donkey serum. Sections were incubated overnight at 4 °C in primary antibody and 2 h in secondary antibody at room temperature. The following primary antibodies were used: rabbit anti-tRFP (1:1000; Evrogen; AB234), rabbit anti-melanopsin (1:1000; Advanced Targeting Systems; AB-N39), goat anti-ChAT (1:100; Millipore; AB144P) and mouse anti-Goα (1:1000; Millipore; mab3073). Secondary antibodies were always from donkey and either conjugated to Alexa 488 or Alexa 594 (1:400; Invitrogen). Alexa 488 conjugated to streptavidin was used to visualize cells injected with biocytin during patch-clamp experiments (1:400; Invitrogen; S-11223). Nuclei were stained with 10 µg/ml DAPI (Roche). Micrographs were taken on a Zeiss LSM 880. Processing of image stacks was done using ImageJ (Rasband WS, United States National Institutes of Health, Bethesda, Maryland, US).

**Multi-electrode array recordings**. Retinas were placed on multi-electrode arrays (60MEA200/30iR-Ti; Multi Channel Systems MCS GmbH) coated with Corning™ Cell-Tak Cell and Tissue Adhesive (Corning) with the ganglion cells facing towards the electrodes. The MEA was placed into the MEA recording device (MEA2100-System; Multi Channel Systems MCS GmbH) positioned on a stage of a Zeiss Axioskop coupled to a pE2 light stimulator (precisExcite, CoolLED, Andover, United Kingdom) connected to an oscilloscope (Tektronix TDS210) and signal generator (ELV TIG7000). Perfusion with oxygenated Ames' medium (Sigma-Aldrich) was maintained at 5 ml × min$^{-1}$. Temperature was maintained at 34 °C. After placement of the MEA into the recording device, the retina was perfused with oxygenated Ames medium for 30 min in darkness. Light stimulation (unless stated otherwise: 465 nm, $5 \times 10^{14}$ photons cm$^{-2}$ s$^{-1}$) was delivered through a 5× objective positioned above the MEA recording device. Recorded signals were collected, amplified, and digitized at 25 kHz using MCRack software (Multi Channel Systems MCS GmbH). Filtering was done using a 2nd order Butterworth high-pass filter with a cut-off frequency of 200 Hz. Action potentials were defined as electrical activity below 3.5–5 SDs of baseline activity and set automatically for each recording electrode based on the baseline noise. Subsequently, spike cutouts recorded by each electrode were sorted into single cell traces using Offline Sorter (Plexon). Time points of single cell spike occurrences were extracted from the software for offline analysis using Matlab (MathWorks).

Each recording, unless stated otherwise, consisted of five consecutive light stimulations. Time points of spike occurrences were extracted around stimulation periods, time binned and averaged. Cells were defined as light responsive using two parameters, with the requirement of fulfilling both: (i) Treshold (TR) defined as a change in firing rate (baseline + 5*SD), or at least 40 Hz in case of cells with no basal activity, between the average frequency prior to light stimulation and at least 1 time bin during or after the light flash and (ii) using light response index (LRI). We defined as LRI = (maximal firing rate during/after stimulus—average firing rate before light stimulus)/(maximal firing rate during/after stimulus + average firing rate before light stimulus). Only cells with the respective LRI > 0.2 and TR crossing were considered light responsive. Parameters of response properties of light responsive cells (onset of spiking, peak of response, response bias (difference in spiking during and after light stimulation), duration of undisturbed response

(number of consecutive time bins above TR from the onset of the response) and time difference between onset of response and its peak) were used for spectral clustering. Number of clusters was determined via inspection of the Eigengap property and inbuild Davies–Bouldin evaluation. Given the fact that spontaneous firing is higher in *rd1* mice as well as potentially contaminated by upregulation of slow firing iPRGC M1-type responses, spiking frequencies below TR were fixed at TR value in the heat map figures.

For dose-response curve, isolated retinas were kept in complete darkness for 20 min and then tested in a series of brief (500 ms) flashes of blue light (470 nm) at 120 s intervals and over a range of intensities. Retinas were then adapted for 15 min to a moderate indoor light level (light-adapted; white light, 100 $\mu$W cm$^{-2}$) and retested as described above. Dose-response curve analysis was performed similarly to the response clustering analysis described above. To accommodate for the absence of the possibility of averaging (as each light intensity in given state was presented only once) the TR metric was adjusted to: baseline + 3*SD, or 40 Hz in cells without basal activity. Cells were considered light responsive when response were observed at the 2 highest light intensities in dark-adapted cells as well as highest light intensity in light-adapted cells. This way we compensated for possible false-positive responses.

A response was defined as:
[peak firing rate after light stimulation] − [mean firing rate before stimulation] (in case of negative values this number was fixed at 0).

Subsequently, responses for each light intensity were normalized to their minima and plotted as normalized averages ± s.e.m. to their corresponding light intensities in Log10 scale. Half-saturations (EC$_{50}$), hill slope and maximal response (Vmax) were calculated by fitting a Hill equation using Prism (GraphPad). The values shown in the plots in Fig. 6c are normalized to Vmax.

**In vivo electrophysiological recordings from the visual cortex V1**. Mice were anesthetized by an intraperitoneal (IP) injection of ketamine and xylazine and subsequently mounted in a stereotaxic device (David Kopf, USA). The pupil of the stimulated eye was dilated with a drop of 10 mg/ml atropine sulfate (Théa Pharma). Subsequently, a thin layer VISCOTEARS eye gel (Bausch & Lomb Swiss AG) was applied to the eyes to prevent drying and allowing clear optical transmission. After a scalp incision, the fascia was removed from the surface of the skull and a craniotomy ~ 1 mm in diameter was made over the left occipital cortex. The electrode was inserted 1.5–2.5 mm lateral and 0.1–0.5 mm rostral to lambda into V1. The depth of the tungsten electrode was zeroed at the pial surface after penetrating the dura. The tungsten microelectrodes (WE3PT30.01A5, Microprobes) had an impedance of 9–15 k$\Omega$ and were guided diagonally through the V1 to reach layer 4 using a micromanipulator and stereomicroscope (Amscope). The temperature of the animal was maintained at 37.5 °C by an animal temperature controller (ATC 2000, World Precision Instruments, Germany). After insertion of the electrode, mice were dark adapted for 30 min. For the light stimulation paradigms we used Samsung Sync master 940b LCD display (60 Hz refresh rate, luminance $5 \times 10^{13}$ photons/cm$^2$/s) located 20 cm away from the eye. Signals were acquired, amplified, and digitized at 20 kHz with the RHD 2000 system from Intan Technologies (version 1.5.2). Recordings were analyzed offline using Matlab. Visually evoked potentials (VEPs) were extracted by down-sampling the raw signal to 1 kHz. Signals were averaged and overlayed for different experimental groups, with the selection of only VEP-exhibiting traces. Light-adapted experiments had the periods of darkness between stimulations exchanged for periods of light stimulation (white light at 100 $\mu$W cm$^{-2}$). Bar plots are shown as means ± SD, exemplar traces as means ± s.e.m.

**Optomotor reflex measurement**. Prior to treatment, *rd1* litters were randomly divided into treatment and control groups. We employed Striatech's LCD monitor-based virtual automated optomotor system to assess the spatial frequency and contrast thresholds of a specific optomotor behavior in awake, unrestrained mice sitting on an elevated platform (9 cm diameter, 10 cm height). The mice were >p168 of age, when no OMR responses remain in untreated animals[6,28]. The brightness of the screens was adjusted to $5 \times 10^{13}$ photons cm$^{-2}$ s$^{-1}$. Tracking head movements were bilaterally recorded by an infrared-sensitive digital camera and analyzed using the OptoDrum software (Striatech, v.1.2.8). The rotation speed was kept constant at 12°/s, which was shown to elicit an optimum response under photopic conditions[51]. Visual acuity was examined with a stair-step protocol of increasing spatial frequencies (100% contrast) and threshold defined as the highest grating frequency that can evoke animal head tracking[29,52]. Contrast sensitivity, the inverse of the Michelson contrast [C = (I max − I min)/(I max + I min)] was determined analogously at 0.125 cyc/deg spatial frequency, where injected rd1 mice performed best (Supplementary Fig. 4). A mouse, which was not able to perform at 100% Michelson contrast was considered blind. Data was obtained by three independent experimenters blinded to the identity of the injected optogene on 3–5 separate measurement days. Tracking thresholds were automatically evaluated by the OMR software.

**Open-field box experiments**. Experiments were performed in a two-compartment shuttle-box (27.5 × 30.5 × 35 cm) connected by a 6 × 5.5 cm gate (Fig. 7a) with rd1 mice >p280 of age, when no cortical responses to pattern stimuli can be observed[6]. We

used a custom-made, two-compartment Plexiglass shuttle-box, each compartment measuring 27.5 cm × 30.5 cm × 35 cm and connected by a 6 × 5.5 cm gate (Fig. 7a). The floors were equipped with removable shock grids (Med Associates Inc, Model VFC-005A) to administer electric foot shocks during cued fear conditioning on days 2–4 (Fig. 7a). At the two opposite short walls of the setup, computer screens (SAMSUNG SyncMaster 940B, Type: GH19PS, width: 38 cm, height: 30.5 cm) were mounted run by a script on Noldus software (EthoVision XT, Version 11), providing equiluminescent gray or moving gratings of 0.35 cyc/deg spatial frequency (100% contrast, speed 12 deg/s). The other sides of the compartments were covered with reflective foil to display reflections of the displayed stimulus. Under the boxes, an IR-illumination pad (Noldus) was placed to homogeneously illuminate the setup from below. Behavior was video recorded with an IR-sensitive BASLER camera (Model acA1300-80GMnir). The experiment consisted of four sessions on four consecutive days (8:00–12:00 am) as described in Fig. 7a. Day 1 (Habituation): Prior to the experiment, the setup was cleaned with 70% ethanol. Both screens were set to equiluminescent gray, and the mouse left to explore for 30 min. If a mouse failed to explore, e.g., to spend approximately equal time on each side of the shuttle box, it was excluded from the experiment. Days 2 and 3 (conditioning): Prior to the experiment, the setup was cleaned with 70% ethanol. Both screens were set to present equiluminescent gray. The mouse was randomly placed into either side of the shuttle-box and left to explore for 5 min to habituate. At 5 min into the experiment, on the side of the setup where the mouse resided at that time the screen was set to present the moving stripe pattern for 1 min paired with foot shocks of 0.6 mA (500 ms every 5 s). Three visual cue sessions paired with foot shocks were performed over a 15 min conditioning period, each followed by 4 min in which both screens displayed equiluminescent gray and no shocks were administered. Day 4 (Evaluation): Prior to the experiment, the setup was cleaned with isopropanol, and the shock grids covered with Plexiglass sheets to prevent environmental cues. After a 5 min habituation period, the screen on the side where the mouse was located at the time switched to the stripe pattern, while the other screen in the second compartment continued to display equiluminescent gray. Aversive behavior (freezing, residing in the opposite compartment, jumping, or tail rattling) was manually scored as cumulative time in for 1 min before the onset of the pattern stimulus (baseline) and 1 min during stimulus presentation.

**Statistics and reproducibility**. For patch-clamp recordings, Student's t-tests was used. For MEA and cortical responses, either unpaired 2-tailed Student's t-test or 2-tailed Kolmogorov–Smirnov test with Bonferroni correction was used. If not stated otherwise, mean ± SD are given.

Behavioral data was analyzed with a one-way ANOVA for multiple comparisons with post hoc analysis with Tukey's honestly significant difference (HSD) of means in R v.3.6.0. Assumptions of normality were not rejected by the Shapiro–Wilk normality test and homogeneity of variance was tested with the Levene's or the Bartlett test. Data is depicted as boxplots indicating the median ± standard deviation.

In graphs, the significance levels are indicated as $p \leq 0.05$, $p \leq 0.01$, and $p \leq 0.001$, unless stated otherwise.

**Reporting summary**. Further information on research design is available in the Nature Research Reporting Summary linked to this article.

## Data availability

The datasets used and/or analyzed during the current study are available from the corresponding author upon reasonable request. Datasets used for the creation of the main figures in this manuscript are included as Supplementary Data 1. Sequence data that support the findings of this study have been deposited in GenBank with the accession codes MQ072285.1 {Mela(CTmGluR6)} and MQ072299.1 {Mela(CT+L3mGluR6)}.Plasmids sequence data used in this study have been deposited at Addgene, under IDs 191343 {Mela(CT+IL3mGluR6)} and 191344 {Mela(CTmGluR6)}.

## Code availability

The codes/algorithms used during the current study are available from the corresponding author upon reasonable request.

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

## Acknowledgements

We would like to thank Matti Zbinden and Simon Hostettler to help perform behavioral experiments, Günther Zeck and Ludovic Mure for fruitful advice and discussions on retinal multi-electrode array recordings, Anwesha Bhattacharyya for advice on cortical recordings, Giulia Schilardi for support in the immunohistochemistry and Christian Dellenbach and Hans Ruchti for electronic support. We also thank Sabine Schneider for her excellent support in cloning, viral packaging, and histology and Michael Känzig for taking care of the animals. We would also like to thank Dr. Beatriz Vidondo for fruitful discussions and her support in the statistical analyses. This research was funded by the Swiss National Science Foundation (31003A_152807 and 31003A_176065), the Bertarelli Foundation (Catalyst fund, project BCL7O2), the Haag-Streit Holding AG, and Arctos Medical AG.

## Author contributions

Conceptualization, S.K.; methodology, S.K., M.v.W., and J.K.; software, J.K.; formal analysis, S.K., M.v.W., and J.K.; investigation, M.v.W., J.K., and N.S.; resources, S.K.; writing original draft preparation, S.K.; writing—review and editing, S.K., M.v.W., and J.K.; supervision, S.K.; project administration, S.K.; funding acquisition, S.K.

## Competing interests

S.K. and M.v.W. are inventors on the patent application P180999CH00 that discloses the chimeric proteins Me-la(CTmGluR6) and Mela(CT+IL3mGluR6). All other authors declare no competing interests.
