## [Peer Review File · Communications Biology]

Reviewers' comments:

Reviewers #1/#2 (Remarks to the Author):

Kralik and colleagues use techniques to express new light-sensitive proteins in retinal bipolar cells, which are the cells that receive glutamatergic input from photoreceptors, communicating these signals to retinal ganglion cells and amacrine cells. In diseases that impact photoreceptors, bipolar cells are a therapeutic target for vision restoration as they remain intact and are present at high densities, which might allow better visual acuity than strategies targeting retinal ganglion cells, at least outside the fovea. This manuscript will be impactful and influential as it convincingly illustrates the importance of creating chimeras of different visual proteins to maximize the therapeutic potential of vision restoration.

The use of GPCR opsins allows amplification, light adaptation, and improved photosensitivity, expanding the functional range of vision restoration beyond those provided with other optogenetic proteins such as channelrhodopsin. They use the following previously established approaches many of which were established by the Kleinlogel group: 1) opsins were localized to bipolar cells (Kralik et al. 2021). 2) The use of light-sensitive chimeras (GPCR melanopsin + mGluR6) and native signaling pathways to restore fast photosensitivity (van Wyk et al. 2015). 3) The restoration of visual acuity in rd1 mice using mammalian opsins or chimeric opsins. They build on this work in the following key ways. 1) They describe two additional melanopsin based chimeras and compare them with previously established opsin approaches. 2) They characterize the photoresponses of their best performing candidate, Mela(CTmGluR6), and identify that response diversity is recorded at the level of the RGCs. 3) Mela(CTmGluR6) can restore behavioral vision to late stage rd1 mice.

The novel finding here is the use of a melanopsin chimera that can avoid bleaching and regenerate with longer wavelength light. There is an impressive amount of work put into this paper and it has some very impressive and difficult experiments, which include recording from bipolar cells in wholemount preparations of retina and in dissociated preparations, and behavior. I have some concerns about the following experiments and suggest they are addressed with either further experiments or a careful description in the text:

- Chimeras were based on the melanopsin protein and replaced the native opsin carboxyl termini with that of the mGluR6 to accelerate the kinetics of melanopsin. However, there is no comparison made between the new chimera opsins and native melanopsin. Vision restoration using mammalian melanopsin has restored functional vision in rd1 mice (De Silva et al. 2017, Lin et al. 2008). Furthermore, melanopsin kinetics (especially the length of the response) are influenced by the amount of melanopsin expression and brightness of light exposure. Early characterization (Tau ON Tau OFF etc – fig 1) of the new chimera opsins should be compared with the complete melanopsin protein under the same lighting conditions. Experiments in HEK cells will be sufficient. The authors might choose to include a discussion
- A particularly impressive and exciting result is the diversity of photoresponses restored using a promoter that targets both ON and OFF bipolar cells. This richer signal at the level of the RGCs has the capacity to restore more detailed vision and provides the first truly convincing motivation for restoring photosensitivity up stream of the RGCs. However, cell specific promoters injected at high titers are prone to off-target expression. Given the sensitivity of the chimeric opsin, even small amounts of off target expression could have functional significance. Therefore, the diversity in responses might be attributed to transfection, (and therefore optogenetic activation) of BCs and RGCs together. To address this, the authors could perform recordings before and after pharmacological isolation of RGCs to show that ON and OFF transient or sustained responses etc. are driven from inner retinal signaling (i.e are photoresponses abolished with excitatory synaptic blockers?).
- Equally as interesting and particularly relevant to the novelty of this work is the tri-stability of melanopsin (fig 1G) which avoids bleaching. Why then did the authors not use the same orange background and blue light stimuli to drive spike responses in the RGCs?

- Figure 5A and 5B. The stratification images are not convincing and are not quantified, particular in 5A where the ChAT somas are almost touching. It is impossible to conclude from these data that these RGCs have their dendrites in the OFF layer.

Minor suggestions:

It would be useful to have a diagram at the start of the manuscript with a cell and ion channels and intracellular pathways showing the cascade for each construct in HEK cells and in ON bipolar cells. This could be easily made in BioRender.

A diagram/image of the optomotor behavior would be useful in Fig1 before H to help direct readers attention to the behavior underlying the data in H. Similar to later behavior figures.

Were behavioral experiments for the optomotor mice (or other behaviors) performed with blinding (i.e. were experimenters aware of the construct injected) and was the analysis performed blind to the AAV injected? If yes, please state, if no, please also state. The methods mention experiments were performed by three independent observers but fail to mention blinding.

Fig 2A 1_2 should be greyscale as shown in C.

Benjamin Sivyer and Michael H Berry

Reviewer #3 (Remarks to the Author):

In this report, Kralik, Kelinlogel and colleagues present a comprehensive characterisation of the extent of restored retinal signalling and behavioural vision following the introduction of modified optogenetic tools targeting bipolar cells. The overall strategy is not in itself novel, as there are an increasing number of reports describing optogenetic-mediated restoration of light sensitivity, achieved by variously targeting retinal ganglion cells and bipolar cells, and using a variety of native and non-native opsins. Indeed, the authors note that there are clinical trials in progress for the use of viral vector-mediated expression of channelrhodopsin in the diseased retina with some early indications of improvements in light sensitivity. However, non-native opsins suffer from poor sensitivity, requiring very high light intensity stimuli. Native opsins offer the potential for function within the normal visual range, but it is not clear how well inner retinal neurons will be able to replicate the phototransduction cascade downstream of opsin isomerisation. Moreover, due to limitations of viral vector tropism, the majority of reports to date have targeted retinal ganglion cells, rather than inner retinal neurons, which misses out a significant part of visual processing that takes place in the inner retina.

Here, Kleinlogel and colleagues present interesting data describing novel melanopsin-mGluR6 chimeras, where specific intracellular domains of melanopsin have been partially replaced with that of mGluR6, creating a melanopsin that shows tropism for Gao G-protein signalling pathway, rather than their natural tropism for Gαq. Moreover, they specifically target bipolar cells and present in-depth characterisation of the restored retinal activity following the introduction of one of three melanopsin and one ML cone opsin mGluR6 chimeras (some previously reported).

In general, this is a well written and considered article, with appropriate levels of caution and qualification, with some exceptions. The authors examine functional rescue at all levels from single cell, through retinal, cortical and behavioural assessments. Of note, they examine rescue in older mice, 24wks of age, which is particularly welcome as so many use much earlier time points when there can still be some residual function that complicate interpretation.

I do have a number of points I would like the authors to address:

1. Intro:

- i) P1, l22 - The diseased eye cannot be described as immune-privileged as the normal barriers are often compromised. Given that this is a therapeutically-positioned paper, the authors should make the distinction between healthy and diseased immune status clear and refer to the diseased eye as having partial immune privilege.
- ii) P1, l24 - optogenetics doesn't create "replacement photoreceptors" out of inner retinal cells, phrase that implies a trans-differentiation or regeneration mechanism - it makes inner retinal cells light-sensitive. Since a significant concern about this approach is whether such cells can faithfully replicate the phototransduction cascade it is inaccurate to claim they are now photoreceptors. Again, please take careful with the 'spin'.
- iii) P3, l138 - reference missing for "previously reported long term functional preservation..."

2. Re OMR data, Figure 1:

- i) It is surprising that the contrast sensitivity of the Mela (CTmGluR6) is almost identical to WT. Since the authors have data for both visual acuity and contrast, it would be helpful to see them both on the same graph to see if the relationship for each chimera is maintained across all tested values when compared to Opto-mGluR6 or Opn1MW, and if not, in which direction is it shifted - only contrast, only acuity, both?
- ii) Please can the authors clarify whether the n numbers (OMR acuity and contrast thresholds) are the average of the two eyes/mouse or whether each eye is listed as an individual value. This should be noted since if the latter, these are not fully independent values. Indeed, it is perhaps surprising that the authors did not make use on internal controls and inject one eye for OMR tests. I understand the need to inject both for the behavioural tests and higher visual area assessments, but it appears that different cohorts were used for these anyway (so the total number of animals involved would not be notably increased)?

3. Single cell recordings, Figure 3:

- i) Panels B&C - In the text the authors say that these two graphs show a "diverse array of intrinsic processing of the mela mGluR6 signal at the level of the OBC". However, panel B is an average of just 6 cells and panel C shows a single cell showing the opposing response. Please rephrase this sentence to reflect the size of the data set.
- ii) Panels E&F. The authors suggest that the cells recorded in the intact retina have more robust and faster recovery times than isolated transfected cells. This is a not an unexpected assertion but needs some quantification of all the cells recorded under both conditions to support it.
- iii) Panel G compares a treated mela-mGluR6 rd1 bipolar cell with an untreated WT BC, where rod/cone input is still functional. It is not entirely clear what this comparison is adding to our understanding? Can the authors explain their rationale and/or justify its inclusion?

4. Extracellular population recordings, Figure 4:

- i) In the figure legend, the authors say that they can identify all "canonical receptive field types". They also claim that they "confirm that activation of OBCs restores inner retinal signaling and with that, RGC receptive-field diversity, an important building block for high quality vision." It is not clear to me how the authors have determined receptive fields from an apparently uniform (non-spatially defined) stimulus? If we understand correctly, they are describing the light response profiles of the RGCs, rather than defining actual RGC classification. We would suggest rephrasing to restoring a "diversity of RGC light-responses".
- ii) In addition, the authors do not discuss the melanopsin response and how they identify it i.e. how do they know if it is an endogenous melanopsin response, or a slow response from their mela-mGluR6 chimeras?
- iii) We assume that the authors only use L-AP4 to block the ON pathway, as it is the ON BCs that they are trying to target. However, they go on to compare both ON and OFF responses in this figure, so one might expect them to also block the OFF pathway, using DNQX and D-AP5, to investigate/isolate

any native photoreceptor responses in WT. If DNQX/D-AP5 has no effect on the response, it would at least provide further support of their hypothesis for different pathways being used.

The authors also mention amacrine circuitry a number of times, but do not try and manipulate it. This would be interesting to explore to see if they can test how they generate the different ON and OFF responses, although this may be considered beyond the scope of the current study.

iv) Panel B of this figure – there appear to be error bars shown for OFF-RGCs BL6_mela but not the other cohorts?

v) Whilst this may be considered a personal preference, displaying data as a fraction of a whole graph is preferable over a pie chart, since the former allows one to better compare the proportional shifts between groups (WT vs rd1 Mela(CTmGluR6, in this case). We would also suggest keeping the number of melanopsin cells in these comparisons, as we know that it should be consistent and not change between RGCs due to degeneration (approx. 6-8%).

The labelling of these pie charts is messy and could be improved.

Figure 6:

i) The % of responsive cells in the WT MEA recordings is surprisingly low at 60%. We would expect these to be more in the range of 90-100%. Can the authors comment on a potential explanation for this? The SD error bars also do not look like they should be NS compared to the treated mice. Perhaps a violin plot would be a better way of representing the true spread of the data points for all three groups?

ii) This may be a pdf conversion issue, but there seem to be 6 titles (ON-transient, ON-sustained, biphasic ON-OFF, OFF-sustained and ipRGCs like) but only 5 graphs under them? There doesn't appear to be a graph for the ipRGC-like.

In the same panel, it is not clear how the ON sustained and OFF sustained have been differentiated from the Melanopsin-like cells and how much overlap there could be?

iii) The justification for use of a sigmoid curve to model the light- and dark-adapted intensity data points is unclear, as neither plot plateaus at a maximum and it is a significant assumption to state that they would. To describe a shift, the authors would need to reach maximum responses for both in order to fit the curves as they have done. The 'gold' data points look like they max out at 0.5 of the normalised response so it is unclear how they have plotted the curve shown to this particular data.

iv) Please take care to check figures for overlapping text/panels e.g. text/scale bar in ON-OFF in fig6B, and panel A overlapping with panel B in same figure.

Figure 7:

i) In Panel E can the authors comment on why the error bars are so large for mela-mGluR6? The data in this panel would be better represented in a violin plot, or at the very least showing data points overlaid (as used in F).

ii) Fig. 7B – for ease of interpretation, please include the strain (WT, rd1 etc), as done for panel C.

Methods:

Please state the age at which the mice were injected in the main Methods, not just in the Supplementary. This is key to the potential extent of rescue.

Ivo Lieberam, PhD
Editorial Board Member
Communications Biology

Bern, June 10, 2022

Dear Dr. Lieberam, Dear reviewers,

Thank you very much for the constructive and positive comments and the opportunity to review our manuscript. We have now performed additional experiments and addressed all points raised in the revised version of the manuscript.

Referees #1/#2

Question 1: How does native melanopsin compare to the melanopsin/mGluR6 chimera used in this study? A basic in vitro study in HEK cells could help address this point.

Response 1: *We have addressed this question two ways: (1) data for native melanopsin was added to figures 1b-e and (2) a detailed comparison of the Tau(OFF) kinetics of native melanopsin with Mela(CTmGluR6) was added as supplementary figure S2. In particular, we now show a comparable native melanopsin trace in panel 1b, and included the values of native melanopsin in panels 1c-e quantifying amplitude, Tau(ON) and Tau(OFF) values of GIRK currents triggered by melanopsin in relation to other opsins including Mela(CTmGluR6). While there is a significant increase in current density when the C-terminus of native melanopsin is replaced by that of mGluR6, there is no significant changes in the ON-kinetics and the primary OFF-kinetics time constant (Tau(OFF)_1). The reviewer is correct that melanopsin kinetics is influenced by the amount of melanopsin expressed and the time and intensity of light exposure. The light intensity and stimulus length was always kept the same in HEK293 experiments (5 sec; 470nm; 10^{14} photons/cm²/s), which is now stated in the respective methods section. Expression levels and opsin localization were determined by mKate2 fluorescence mediated by the fusion proteins. In the new Suppl. Fig. S2 we compare the blue-light elicited responses of a Mela(CTmGluR6)- and a Mela(WT)-expressing HEK293-GIRK cell with similar mKate2 fluorescence comparable to that shown in Fig. 1a. This new figure nicely shows that the CT-replacement of Melanopsin by mGluR6 modifies the off-kinetics, besides an acceleration also mediating full cessation of the GIRK current, which is not the case for native melanopsin. Besides updating figures 1b-e and adding suppl. Fig. S2, we also adapted the Results text and Fig. 1 legend accordingly.*

Question 2: a) How bipolar cell-specific is the promoter used in this study? Do high titres of AAV vector lead to ectopic expression in retinal ganglion cells?

Response 2 a: *We have fully characterised the cell-type specificity of the 770En_454P(hGRM6) promoter in a previous study published in 2020 (Hulliger et al., 2020, <https://doi.org/10.1016/j.omtm.2020.03.003>). The promoter employed is highly specific for ON-bipolar cells, but, as alluded to by the reviewer, has some off-target expression at the injection site*

where AAV load is high. However, off-target optogene expression (published previously and demonstrated in Figs. 2B&C) is very low.

Question 2 b: Even small amounts of off target expression could have functional significance. Therefore, the diversity in responses might be attributed to transfection, (and therefore optogenetic activation) of BCs and RGCs together. To address this, the authors could perform recordings before and after pharmacological isolation of RGCs to show that ON and OFF transient or sustained responses etc. are driven from inner retinal signalling (i.e are photoresponses abolished with excitatory synaptic blockers?).

Response 2b: This is indeed an important point raised by the reviewer, which we addressed by recording from 8 additional rd1 mice injected with Mela(CTmGluR6). We show that weak off-target expression of Mela(CT mGluR6) in retinal ganglion cells makes no contribution to the diversity of the RGC responses mediated by Mela(CT mGluR6) expressed in ON-bipolar cells. These new results are summarized at the end of section 2.4. and shown in the new Supplementary Fig. S11. In line with the sparsity of off-target expression observed (as shown in Fig. 2b in the representative immunohistochemically labelled retinal slice) we only found three retinal ganglion cells within 16 whole mount retinas (from the 8 bilaterally injected rd1 mice) with detectable TurboFP635 expression to record light-signals. We used a standard cocktail of CNQX and D-AP5 to block input from inner retinal cells. The light responses measured under these conditions of Mela(CT mGluR6)-expressing RGCs mirrored responses of ipRGCs and were slow with a late peak in firing frequency after the offset of the light stimulus. These response types were routinely classified as ipRGCs (see line 253ff of the manuscript) and did not contribute to the reported diversity of Mela(CT mGluR6) mediated RGC responses. We also labelled the cells after recording with biocytin to make sure we did not record from M1 ipRGCs with intrinsic melanopsin responses, but from RGCs that became light-responsive as a consequence of the expression of Mela(CT mGluR6). The morphologies of the labelled RGC cells were not of the typical M1 ipRGC type, but typically bi-stratified confirming that light responses were not mediated by endogenously expressed melanopsin, but introduced through ectopic expression of Mela(CT mGluR6). We therefore conclude that (1) off-target expression of Mela(CT mGluR6) in RGCs is extremely low with the hGRM6 promoter and AAV2(7m8) capsid combination used, (2) that ectopic expression of Mela(CTmGluR6) in RGCs did not contribute to the diversity of light responses reported as being mediated by Mela(CTmGluR6) expressed in ON-bipolar cells, (3) that RGCs possess Gi-mediated intracellular pathways that can be hijacked by the Mela(CTmGluR6) chimera and 4) that due to potential off-target signalling, the off-target expression rates have to be kept low in a bipolar cell targeted approach. These experiments were added to p.220ff in the Results section and in the last paragraph of the discussion.

Question 3: The tri-stability of melanopsin appears to effectively avoid photobleaching. Why then did the authors not use the same orange background and blue light stimuli to drive spike responses in the RGCs?

Response 3: Illumination with an orange backlight mainly accelerates the kinetics of Mela(CT mGluR6). Light-switching of an opsin protein is extremely rapid, typically in the microsecond range, whereas response kinetics of retinal neurons are rather slow and lie in the 10-100 ms range. In the On-bipolar cells, therefore, the kinetically limiting factor is not the switching time of the bi-stable opsin, but the release kinetics of beta/gamma from Galpha(o) triggering TRPM1 channel closure as well as

cell-specific arrestin and RGS kinetics to terminate G-protein signalling. In the RGCs, response kinetics and response diversity is in our opinion a consequence of ON- and OFF-channel input from the inner retina as well as RGC receptive field organisation. Important in this respect is the proven functionality of the All amacrine cell circuitry, which transfers the light-response from the optogenetically targeted ON-channel also into the OFF-channel.

The tri-stability of melanopsin on the other hand, as shown in previous studies and demonstrated here (Fig. 1g) confirms a closed-loop-photocycle. In other words, melanopsin retains the light-isomerized chromophore, which can then be re-isomerised either by a “slow” intrinsic dark-isomerase, or by orange light. This property is important in the context of vision restoration because it makes melanopsin independent of the visual cycle (enzymatic recycling of 11-cis-retinal) located in the RPE, which is also largely compromised in photoreceptor dystrophies. OPN1MW for example, a monostable pigment, does not possess this important property, which may limit the outcome of vision restoration when targeting retinas lacking a functional RPE. It is this “bleach resistance” of melanopsin we intended to demonstrate.

The reviewer may also refer to the incomplete recovery of the Mela(CTmGluR6) response in Fig. 1f upon repeated illumination without a backlight. We indeed observed a reduction of the response amplitude during the first light pulses in HEK293 cells, which then, however, stabilized at approximately 35-40% of the initial response. We consider this reduction not an effect of bleaching, but a potential consequence of a shift in balance towards a non-conducting state within the photocycle. Importantly, this reduction was NOT observed *ex vivo* when recording from bipolar cells in retinal whole mounts (see Fig. 3f) and the response amplitude to a blue light stimulus without backlight sufficed to trigger functionally relevant responses in all *ex vivo* and *in vivo* experiments. We therefore considered an orange backlight for *in vivo* applications not necessary, in particular since this would have also complicated our experimental illumination paradigms. Further, the use of concomitant orange backlight would not allow non-aided natural light stimulation in a potential future patient; i.e. the visual stimulus would need to be adapted and displayed by biomimetic goggles as in the ongoing GenSight trial. We eluded to this in line 391 of the manuscript.

Question 4: Please provide better-quality figures for the stratification images, and quantify the data in Fig.5A/B.

Response 4: We agree that the original Z-projections were not very clear. We have now updated Figs. 5A&B with Z-projections that separate the somata of choline acetyltransferase (ChAT) expressing retinal amacrine cells better to show dendritic stratification more clearly. We added quantifications to the Results section in line 254ff and to the legend of Table S1 (in red).

Questions 5/6: Please add diagrams at the start of the manuscript which shows ion channels/intracellular pathways, and illustrate optomotor behaviour.

Response 5/6: We now made a new supplementary figure (Fig. S1) that shows the ion channels/intracellular pathways with one of our constructs (Mela(CTmGluR6)) in both, HEK293-GIRK cells and retinal ON-bipolar cells. Illustration of the optomotor behaviour was now added as new panel h in Fig. 1.

Question 7: Please provide more details on the behavioural experiments (blinding etc.).

Response 7: The according methods have been updated in section 4.7. The experimenters were blinded to the identity of the injected optogene and the tracking threshold was automatically evaluated by the OMR software.

Question 8: Fig 2A 1_2 should be greyscale as shown in C.

Response 8: *This has now been adapted.*

Referee #3:

1. Intro:

i) P1, l22 - The diseased eye cannot be described as immune-privileged as the normal barriers are often compromised. Given that this is a therapeutically-positioned paper, the authors should make the distinction between healthy and diseased immune status clear and refer to the diseased eye as having partial immune privilege.

Response 1i: *This was now changed in line 22.*

ii) P1, l24 – optogenetics doesn't create "replacement photoreceptors" out of inner retinal cells, phrase that implies a trans-differentiation or regeneration mechanism - it makes inner retinal cells light-sensitive. Since a significant concern about this approach is whether such cells can faithfully replicate the phototransduction cascade it is inaccurate to claim they are now photoreceptors. Again, please take careful with the 'spin'.

Response 1ii: *This was now changed in line 24 to "...renders remaining inner retinal cells light sensitive".*

iii) P3, l138 – reference missing for "previously reported long term functional preservation..."

Response 1iii: *An appropriate reference was now added (doi: 10.3390/ijms222111515).*

2. Re OMR data, Figure 1:

i) It is surprising that the contrast sensitivity of the Mela (CTmGluR6) is almost identical to WT. Since the authors have data for both visual acuity and contrast, it would be helpful to see them both on the same graph to see if the relationship for each chimera is maintained across all tested values when compared to Opto-mGluR6 or Opn1MW, and if not, in which direction is it shifted - only contrast, only acuity, both?

Response 2i: *Correlation plots were now added in the new supplementary figure S5. All experimental groups, except OPN1MW(CTmGluR6) show a significant positive correlation similar to WT C57BL/6 mice between restored contrast sensitivity and restored visual acuity. OPN1MW(CTmGluR6) animals (d, n=5), however, did not exhibit significant positive correlations. The lack of significance in OPN1MW(CTmGluR6) treated rd1 mice could be due to the limited sample size.*

ii) Please can the authors clarify whether the n numbers (OMR acuity and contrast thresholds) are the average of the two eyes/mouse or whether each eye is listed as an individual value. This should be noted since if the latter, these are not fully independent values. Indeed, it is perhaps surprising that the authors did not make use on internal controls and inject one eye for OMR tests. I understand the need to inject both for the behavioural tests and higher visual area assessments, but it appears that different cohorts were used for these anyway (so the total number of animals involved would not be notably increased)?

Response 2ii: *The n numbers represent animals and not individual eyes. This has now been clarified in the manuscript text. The OMR was the first screening, and the same cohort of mice was indeed*

used for the other experiments (open-field box, V1-recording, patch-clamp, MEA) to allow intra-individual comparison of performance. Therefore we performed bilateral injections as already noted by the reviewer.

3. Single cell recordings, Figure 3:

i) Panels B&C - In the text the authors say that these two graphs show a “diverse array of intrinsic processing of the mela mGluR6 signal at the level of the OBC”. However, panel B is an average of just 6 cells and panel C shows a single cell showing the opposing response. Please rephrase this sentence to reflect the size of the data set.

Response 3i: *This was now rephrased in line 185ff, thank you for the comment.*

ii) Panels E&F. The authors suggest that the cells recorded in the intact retina have more robust and faster recovery times than isolated transfected cells. This is a not an unexpected assertion but needs some quantification of all the cells recorded under both conditions to support it.

Response 3ii: *The quantifications are given in the manuscript “The OFF response kinetics of rod-type OBCs was significantly accelerated compared to the isolated configuration (Fig. 3e, TauOFF wholemount: 715 ± 350 ms, $n=9$; TauOFF isolated: 1726 ± 985 ms, $n=11$, $p=0.0001$), but no significant difference was found at the level of the ON kinetics (TauON wholemount: 339 ± 164 ms; TauON isolated: 383 ± 202 m, $p=0.303$).” We hope this is what the reviewer is referring to?*

iii) Panel G compares a treated mela-mGluR6 rd1 bipolar cell with an untreated WT BC, where rod/cone input is still functional. It is not entirely clear what this comparison is adding to our understanding? Can the authors explain their rationale and/or justify its inclusion?

Response 3iii: *The rationale behind the inclusion of this panel is to show the opposing mechanism of action during light stimulation in healthy and treated ON-bipolar cells. In healthy tissue light activation of the retina results in depolarization of the ON-bipolar cells due to a decrease of glutamate release from the photoreceptors and consequently a decrease in native mGluR6 activity, opening TRPM1 channels. The situation is the exact opposite in the Mela(CTmGluR6) animals, where the activation of the optogene results in the hyperpolarization of the cell. It is important to show this as it highlights the ability of the Mela(CTmGluR6) chimera to activate the native mGluR6 signalosome. Moreover, it demonstrates that even in late stages of retinal degeneration the ON-bipolar cells have all the necessary components and functionality to convey downstream light signals, re-confirming previously shown data (van Wyk et al., 2015; Kralik & Kleinlogel, 2021). Lastly, this panel answers the frequent question if an optogenetic protein expressed in the cell body of an ON-bipolar cell can promote comparable membrane potential changes as when the far more abundant cone opsin and rhodopsin molecules in the outer segments of photoreceptors are light-activated. We here show that Mela(palm) triggers substantial light-responses in ON-bipolar cells.*

4. Extracellular population recordings, Figure 4:

i) In the figure legend, the authors say that they can identify all “canonical receptive field types”. They also claim that they “confirm that activation of OBCs restores inner retinal signaling and with that, RGC receptive-field diversity, an important building block for high quality vision.” It is not clear to me how the authors have determined receptive fields from an apparently uniform (non-spatially defined) stimulus? If we understand correctly, they are describing the light response profiles of the RGCs, rather than defining actual RGC classification. We would suggest rephrasing to restoring a “diversity of RGC light-responses”.

Response 4i: *Thanks for the valid comment, we have now rephrased accordingly.*

ii) In addition, the authors do not discuss the melanopsin response and how they identify it i.e. how do they know if it is an endogenous melanopsin response, or a slow response from their mela-mGluR6 chimeras?

Response 4ii: We have now elaborated this point in the text. Melanopsin-type responses are known to increase in rd1 mice with progressing degeneration (doi: 10.3389/fnins.2020.00320), which we identified in treated rd1 mice by comparison with untreated rd1 controls (Supplementary Fig. S10B). In our analysis we focused exclusively on the non-ipRGC like responses that were only found in Mela(CTmGluR6)-treated rd1 retinas. In addition, we have now for the revised version recorded from Mela(CT mGluR6) expressing RGCs under pharmacological block (D-AP5 and CNQX). Responses mediated by Mela(CT mGluR6) expressed in non-ipRGC RGCs resembles that of melanopsin (see new Suppl. Fig. S11 and line 220ff and 263ff in Results section).

iii) We assume that the authors only use L-AP4 to block the ON pathway, as it is the ON BCs that they are trying to target. However, they go on to compare both ON and OFF responses in this figure, so one might expect them to also block the OFF pathway, using DNQX and D-AP5, to investigate/isolate any native photoreceptor responses in WT. If DNQX/D-AP5 has no effect on the response, it would at least provide further support of their hypothesis for different pathways being used.

Response 4iii: We used L-AP4 AND photoreceptor bleaching by high light intensities. Photoreceptor bleaching in treated C57BL/6 mice expressing the optogene was accomplished with maximal epifluorescence illumination with a 150 W Xenon lamp (5.35×10^{17} photons $s^{-1}cm^{-2}$) for 5 min. This protocol leads to complete bleaching of both rods and cones, as calculated by the equation $F = 1 - \exp(-IPt)$ (Wang & Kefalov, 2009, <https://doi.org/10.1016/j.cub.2009.07.054>) with F the fraction of bleached pigment, I the light intensity (5.3×10^{17} photons $cm^{-2} s^{-1}$), t the time of light exposure (300 s), and P the photoreceptor's photosensitivity ($6.0 \times 10^{-9} \mu m^{-2}$ for cones and $6.2 \times 10^{-9} \mu m^{-2}$ for rods). DNQX/D-AP5 does not only block the photoreceptor-to-OFF-bipolar cell synapse, but also the bipolar cell-to-RGC synapse.

iv) The authors also mention amacrine circuitry a number of times, but do not try and manipulate it. This would be interesting to explore to see if they can test how they generate the different ON and OFF responses, although this may be considered beyond the scope of the current study.

Response iv): Indeed, this would be something interesting for the future, particularly since synaptic re-wiring does occur in the degenerated retina. However, we also consider such investigations out of scope of the current manuscript.

v) Panel B of this figure – there appear to be error bars shown for OFF-RGCs BL6_mela but not the other cohorts?

Response 4v: This was now streamlined, thank you.

vi) Whilst this may be considered a personal preference, displaying data as a fraction of a whole graph is preferable over a pie chart, since the former allows one to better compare the proportional shifts between groups (WT vs rd1 Mela(CTmGluR6), in this case). We would also suggest keeping the number of melanopsin cells in these comparisons, as we know that it should be consistent and not change between RGCs due to degeneration (approx. 6-8%).

Response 4vi: Thank you for the comment. In single-cell patch-clamp RGC recordings, melanopsin-like responses were ignored during an initial test stimulus. For this reason, melanopsin cells are not included in the patch-clamp recordings. The fraction of melanopsin responses are instead shown in the MEA recordings, where they were not excluded. Moreover, inclusion of the ipRGC may be misleading the data interpretation in this particular scenario, as there is indeed an increase of melanopsin-like responses in rd1 mice, observed by us and others (Eleftheriou et al., 2020, doi: 10.3389/fnins.2020.00320.). This increase can climb to approx. 37% in the degenerated rd1 retina.

vii) The labelling of these pie charts is messy and could be improved.

Response 4vii: Thank you, this was now adjusted.

5. Figure 6:

i) The % of responsive cells in the WT MEA recordings is surprisingly low at 60%. We would expect these to be more in the range of 90-100%. Can the authors comment on a potential explanation for this? The SD error bars also do not look like they should be NS compared to the treated mice. Perhaps a violin plot would be a better way of representing the true spread of the data points for all three groups?

Response 5i: To exclude any potential bias we have refrained from removing data from retinal pieces that did not show many light responsive cells, e.g. retinal pieces with little transduction. This is now also stated in the methods section. As suggested by the reviewer, we now added individual data points to the plots shown in Fig. 6a. One C57BL/6 retinal explant seemed to exhibit unusually weak light responses. We therefore confirmed normality of distributions using the Shapiro-Wilk test. We then performed a 2-tailed Student's T-test with Bonferroni correction ($\alpha=0.0166$), with the p-values stated in the manuscript. Additionally, we performed a one-way ANOVA (GraphPad, Prism, $\alpha=0.05$) that confirmed the results: C57BL/6 vs rd1 adjusted p-value=0.003 (significant), C57BL/6 vs rd1_Mela(CTmGluR6) $p=0.1461$ (not significant), rd1_Mela(CTmGluR6) vs rd1 $p=0.0369$ (significant).

ii) This may be a pdf conversion issue, but there seem to be 6 titles (ON-transient, ON-sustained, biphasic ON-OFF, OFF-sustained and ipRGCs like) but only 5 graphs under them? There doesn't appear to be a graph for the ipRGC-like.

Response 5ii: Thank you for pointing this conversion error out, which we have now corrected.

In the same panel, it is not clear how the ON sustained and OFF sustained have been differentiated from the Melanopsin-like cells and how much overlap there could be?

Response 5ii2: ipRGC-like cells in treated animals (Fig. 6B) and in control rd1 animals (Fig. S10D) were identified by a delayed and slow response onset during light stimulation, which peaked long after the end of light stimulation. ON- and OFF-sustained cells do not share this characteristic: they show very transient responses at either light on or light off.

iii) The justification for use of a sigmoid curve to model the light- and dark-adapted intensity data points is unclear, as neither plot plateaus at a maximum and it is a significant assumption to state that they would. To describe a shift, the authors would need to reach maximum responses for both in order to fit the curves as they have done. The 'gold' data points look like they max out at 0.5 of the normalized response so it is unclear how they have plotted the curve shown to this particular data.

Response 5iii: Thank you very much for this attentive observation. The anticipated Hill fit was indeed performed incorrectly. In the revised version, the Hill fit was performed correctly on each individual curve, dark- and light-adapted. In figure 6c, both curves were normalized to V_{max} retrieved from the Hill fit and plotted in the same graph. We have adapted the Methods section 4.5. by adding the following fit description: "Responses for each light intensity were normalized to their minimal response in both, dark- and light-adapted trials and plotted as normalized averages \pm s.e.m. Half-saturations (EC_{50}), Hill slope and maximal responses (V_{max}) were calculated by fitting a Hill equation in GraphPad (Prism) to each individual curve. The values shown in the plots in Fig. 6c are normalized to V_{max} ." Hill fit slopes, V_{max} and EC_{50} values are given in the legend of fig. 6c.

iv) Please take care to check figures for overlapping text/panels e.g. text/scale bar in ON-OFF in fig6B, and panel A overlapping with panel B in same figure.

Response 5iv: Thank you, this was done now.

6. Figure 7:

i) In Panel E can the authors comment on why the error bars are so large for mela-mGluR6? The data in this panel would be better represented in a violin plot, or at the very least showing data points overlaid (as used in F).

Response 6i: We have added single data points as suggested. The large variability in *rd1_Mela(CTmGluR6)* mice is almost surely due to patchy and non-homogenous expression throughout the retina, which is typical for a gene therapy.

ii) Fig. 7B – for ease of interpretation, please include the strain (WT, *rd1* etc), as done for panel C.

Response 6ii: This was done.

7. Methods:

Please state the age at which the mice were injected in the main Methods, not just in the Supplementary. This is key to the potential extent of rescue.

Response 7: This was done.

Reviewers' comments:

Reviewer #1 (Remarks to the Author):

Revision

Kralik and colleagues use techniques to express new light-sensitive proteins in retinal bipolar cells as a potential future gene therapy to restore vision in patients that have lost photoreceptor function. They convincingly demonstrate the use of a melanopsin/mGluR6 chimera to restore light responses to bipolar cells and their downstream retinal ganglion cells in rd1 mice. The authors have made substantial and adequate efforts to address the concerns of the reviewers. I have some minor comments that might improve the supplemental figures.

1. Figure S4. It is not clear what is measured here. For example, what does 10% contrast on this bar graph represent? Contrast that elicits an optomotor response? Please clarify in the legend.
2. Fig S7 Melanopsin antibody is unclear here. Please state "rabbit anti-melanopsin in the legend". Methods incorrectly describes ATS N-39 as goat. Please correct this.
3. Figure S9. Instead of labeling the ON-BCs as OFF-BCs in (b) please change to "now OFF-BC" to remove confusion or have a declarative statement at the beginning of the figure that ON-BCs become OFF-BCs.
4. AII AC is purple/magenta not "pink". Remove typos such as "depo-larizes" "subse-quentially"
5. Figure S10. "promi-nent"

Reviewer #2 (Remarks to the Author):

Kralik et al. have addressed all my concerns satisfactorily. I have no further questions or requests. I recommend publication for their manuscript. Congratulations on a thorough study.

Reviewer #3 (Remarks to the Author):

Referee #3:

We thank the authors for addressing all the issues raised. The majority have been satisfactorily resolved and we thank the authors for the efforts made, both experimentally and to the MS, but we do feel there are a couple of issues outstanding:

Point 4. Extracellular population recordings,

Original Reviewer Comment: 4iii) We assume that the authors only use L-AP4 to block the ON pathway, as it is the ON BCs that they are trying to target. However, they go on to compare both ON and OFF responses in this figure, so one might expect them to also block the OFF pathway, using DNQX and D-AP5, to investigate/iso-late any native photoreceptor responses in WT. If DNQX/D-AP5 has no effect on the response, it would at least provide further support of their hypothesis for different pathways being used.

Author Response 4iii: We used L-AP4 AND photoreceptor bleaching by high light intensities. Photoreceptor bleaching in treated C57BL/6 mice expressing the optogene was accomplished with maximal epifluorescence illumination with a 150 W Xenon lamp (5.35×10^{17} photons $s^{-1}cm^{-2}$) for 5 min. This protocol leads to complete bleaching of both rods and cones, as calculated by the equation $F = 1 - \exp(-IPt)$ (Wang & Kefalov, 2009, <https://doi.org/10.1016/j.cub.2009.07.054>) with F the fraction of bleached pigment, I the light intensity (5.3×10^{17} photons $cm^{-2} s^{-1}$), t the time of light exposure (300 s), and P the photoreceptor's photosensitivity ($6.0 \times 10^{-9} \mu m^{-2}$ for cones and $6.2 \times 10^{-9} \mu m^{-2}$ for rods). DNQX/D-AP5 does not only block the photoreceptor-to-OFF-bipolar cell synapse,

but also the bipolar cell-to-RGC synapse.

New Reviewer Comment 4iii: We appreciate the clarification that both pharmacological blockade and light exposure were used here. We note, however, that a paper by Munch and colleagues (DOI: 10.1038/s41467-017-01816-6), which postdates the Wang and Kefalov reference, suggests that rods never fully bleach and start responding at photopic light levels after a short recovery period. We do not think this undermines the authors hypothesis but would suggest the inclusion of this reference as a caveat.

Point 4 iv)

Original Reviewer Comment 4iv): The authors also mention amacrine circuitry a number of times, but do not try and manipulate it. This would be interesting to explore to see if they can test how they generate the different ON and OFF responses, although this may be considered beyond the scope of the current study.

Author Response 4iv): Indeed, this would be something interesting for the future, particularly since synaptic re-wiring does occur in the degenerated retina. However, we also consider such investigations out of scope of the current manuscript.

New Reviewer Comment 4iv): We appreciate the authors response that this may be outside the scope of this study, however, it would be important to conduct these experiments to evaluate the efficacy of targeting bipolar cells with their optogene to know how the signal may spread laterally in the retina, especially as they discuss the lateral spread of native melanopsin responses in the dystrophic retina (native responses increasing from <10% to approximately 37% as stated by Eleftheriou et al., 2020). Might this be happening for the Mela MGlur6 responses too? A brief consideration of this in the discussion would be sufficient.

Point 4 vi)

Original Reviewer Comment 4vi): Whilst this may be considered a personal preference, displaying data as a fraction of a whole graph is preferable over a pie chart, since the former allows one to better compare the proportional shifts between groups (WT vs rd1 Mela(CTmGluR6, in this case). We would also suggest keeping the number of melanopsin cells in these comparisons, as we know that it should be consistent and not change between RGCs due to degeneration (approx. 6-8%).

Author Response 4vi): Thank you for the comment. In single-cell patch-clamp RGC recordings, melanopsin-like responses were ignored during an initial test stimulus. For this reason, melanopsin cells are not included in the patch-clamp recordings. The fraction of melanopsin responses are instead shown in the MEA recordings, where they were not excluded. Moreover, inclusion of the ipRGC may be mis-leading the data interpretation in this particular scenario, as there is indeed an increase of melanopsin-like responses in rd1 mice, observed by us and others (Eleftheriou et al., 2020, doi: 10.3389/fnins.2020.00320.). This increase can climb to approx. 37% in the degenerated rd1 retina.

New Reviewer Comment 4vi): We agree that the number of ipRGC light responses can increase in advanced degeneration through corrupt and functionally reorganised circuitry in the inner retina (as described by Eleftheriou et al., 2020 and others) but, to our knowledge, this has not been shown to be as a result of an increase in the number of melanopsin-expressing RGCs. This would however raise the question of whether the same method of widespread light activity in the inner retina (through electrical or chemical synapses) occurs when testing the efficacy of the Mela MGlur6 optogene in Bipolar cells? (as discussed in response 4iv). We appreciate that this is difficult to factor in for this graph but may be an important sentence to raise in the discussion until future experiments to address this are conducted.

Point 5. Figure 6a:

i) Original Reviewer Comment 5i): The % of responsive cells in the WT MEA recordings is surprisingly low at 60%. We would expect these to be more in the range of 90-100%. Can the authors comment on a potential explanation for this? The SD error bars also do not look like they should be NS compared to the treated mice. Per-haps a violin plot would be a better way of representing the true spread of the data points for all three groups?

Author Response 5i): To exclude any potential bias we have refrained from removing data from retinal pieces that did not show many light responsive cells, e.g. retinal pieces with little transduction. This is now also stated in the methods section. As suggested by the reviewer, we now added individual data points to the plots shown in Fig. 6a. One C57BL/6 retinal explant seemed to exhibit unusually weak light responses. We therefore confirmed normality of distributions using the Shapiro-Wilk test. We then performed a 2-tailed Student's T-test with Bonferroni correction ($\alpha=0.0166$), with the p-values stated in the manuscript. Additionally, we performed a one-way ANOVA (GraphPad, Prism, $\alpha=0.05$) that confirmed the results: C57BL/6 vs rd1 adjusted p-value=0.003 (significant), C57BL/6 vs rd1_Mela(CTmGluR6) $p=0.1461$ (not significant), rd1_Mela(CTmGluR6) vs rd1 $p=0.0369$ (significant).

New Reviewer Comment 5i): We appreciate the clarification and note the inclusion of the individual data points. This does allow the reader to now see the spread of the data. This does raise some further concern, though: 10% light responsive cells in a WT retina is clearly not typical i.e. it is much more likely to be a technical rather than biological basis. The majority of the recordings are up at >85%, which is what would also be expected from the published literature. Since the number of WT recordings is small ($N=6$), the inclusion of that one data point brings the WT average down markedly. It also confers a very large SD to the WT data (almost twice that of the other two groups), which may be the reason that there is then no statistical difference compared to the Opto-treated RD1 cohort. The effect this has is to skew the data and give the impression that the Opto gene therapy is significantly better (i.e. closer to WT) than it actually is. It would seem likely that the removal of this one WT retina would remove the N.S. difference between WT and treated RD1.

We note that the decision to exclude data is a very difficult one and that the authors have used a normality test, but this is of limited strength on this sort of sample size. The inclusion of additional WT retina recordings to bring these control groups closer in N number to the treated numbers ($N=24$) retinas would be an appropriate way to address the issue. This would allow a more accurate assessment of normality, as well as tests for outliers, and would most likely bring the SD within a range similar to that shown in the other groups.

Point 5 iii) Original Reviewer comment: The justification for use of a sigmoid curve to model the light- and dark-adapted intensity data points is unclear, as neither plot plateaus at a maximum and it is a significant assumption to state that they would. To describe a shift, the authors would need to reach maximum responses for both in order to fit the curves as they have done. The 'gold' data points look like they max out at 0.5 of the normalized response so it is unclear how they have plotted the curve shown to this particular data.

Author Response 5iii): Thank you very much for this attentive observation. The anticipated Hill fit was indeed performed incorrectly. In the revised version, the Hill fit was performed correctly on each individual curve, dark- and light-adapted. In figure 6c, both curves were normalized to V_{max} retrieved from the Hill fit and plotted in the same graph. We have adapted the Methods section 4.5. by adding the following fit description: "Responses for each light intensity were normalized to their minimal response in both, dark- and light-adapted trials and plotted as normalized averages \pm s.e.m. Half-saturations (EC_{50}), Hill slope and maximal responses (V_{max}) were calculated by fitting a Hill equation in GraphPad (Prism) to each individual curve. The values shown in the plots in Fig. 6c are nor-

malized to V_{max} ." Hill fit slopes, V_{max} and EC_{50} values are given in the legend of fig. 6c.

New Reviewer Comment 5iii):

We thank the authors for re-examining this, however we are still not certain that the revised presentation is correct mathematically. The graphs indicate that it was not possible to reach the maximum response in either condition, since neither plateaued, which is essential to accurately plot a sigmoidal dose-response curve regardless of whether the data is normalised or not. This would argue against the validity of trying to fit a sigmoid curve with an accurate hillslope. However, we do agree that there is a 1-unit log shift in the responses and so would recommend seeking the advise of a statistician in this instance, to see if there is a more appropriate fit.

Dear Reviewers,

We would again like to thank you for the overall positive feedback and the justified further comments, to which we replied point-by-point below also adding additional data to figures 6a and 6c.

Reviewer #1

The authors have made substantial and adequate efforts to address the concerns of the reviewers. I have some minor comments that might improve the supplemental figures.

1. Figure S4. It is not clear what is measured here. For example, what does 10% contrast on this bar graph represent? Contrast that elicits an optomotor response? Please clarify in the legend.

Authors: This was now clarified in the S4 figure legend. Measured was Michelson contrast, defined as $(L_{max}-L_{min})/(L_{max}+L_{min})$, with L_{max} and L_{min} representing the highest and lowest luminance, respectively.

2. Fig S7. Melanopsin antibody is unclear here. Please state “rabbit anti-melanopsin in the legend”. Methods incorrectly describes ATS N-39 as goat. Please correct this.

Authors: Thank you for pointing out this slip of the pen, which we have corrected.

3. Figure S9. Instead of labeling the ON-BCs as OFF-BCs in (b) please change to “now OFF-BC” to remove confusion or have a declarative statement at the beginning of the figure that ON-BCs become OFF-BCs.

Authors: We have added the declarative statement to the figure legend that “the original ON-BCs become the new OFF-BCs, and the original OFF-BCs become the new ON-BCs” as suggested by the reviewer.

4. Figure S9. All AC is purple/magenta not “pink”. Remove typos such as “depo-larizes” “subsequently”

Authors: This was now corrected.

5. Figure S10. “promi-nent”

This was now corrected.

Reviewer #2

Kralik et al. have addressed all my concerns satisfactorily. I have no further questions or requests. I recommend publication for their manuscript. Congratulations on a thorough study.

Authors: Thank you very much for acknowledging our paper.

Reviewer #3

We thank the authors for addressing all the issues raised. The majority have been satisfactorily resolved and we thank the authors for the efforts made, both experimentally and to the MS, but we do feel there are a couple of issues outstanding:

Point 4. Extracellular population recordings

Original Reviewer Comment: 4iii) We assume that the authors only use L-AP4 to block the ON pathway, as it is the ON BCs that they are trying to target. However, they go on to compare both ON and OFF responses in this figure, so one might expect them to also block the OFF pathway, using DNQX and D-AP5, to investigate/iso-late any native photoreceptor responses in WT. If DNQX/D-AP5 has no effect on the response, it would at least provide further support of their hypothesis for different pathways being used.

Author Response 4iii: We used L-AP4 AND photoreceptor bleaching by high light intensities. Photoreceptor bleaching in treated C57BL/6 mice expressing the optogene was accomplished with maximal epifluorescent illumination with a 150 W Xenon lamp (5.35×10^{17} photons $s^{-1}cm^{-2}$) for 5 min. This protocol leads to complete bleaching of both, rods and cones, as calculated by the equation $F = 1 - \exp(-Ipt)$ (Wang & Kefalov, 2009, <https://doi.org/10.1016/j.cub.2009.07.054>) with F the fraction of bleached pigment, I the light intensity (5.3×10^{17} photons $cm^{-2} s^{-1}$), t the time of light exposure (300 s), and P the photoreceptor's photosensitivity ($6.0 \times 10^{-9} \mu m^{-2}$ for cones and $6.2 \times 10^{-9} \mu m^{-2}$ for rods). DNQX/D-AP5 does not only block the photoreceptor-to-OFF-bipolar cell synapse, but also the bipolar cell-to-RGC synapse.

New Reviewer Comment 4iii: We appreciate the clarification that both pharmacological blockade and light exposure were used here. We note, however, that a paper by Munch and colleagues (DOI: 10.1038/s41467-017-01816-6), which postdates the Wang and Kefalov reference, suggests that rods never fully bleach and start responding at photopic light levels after a short recovery period. We do not think this undermines the authors hypothesis but would suggest the inclusion of this reference as a caveat.

New Author Reply 4iii: Munch and colleagues demonstrated that – against immediate intuition – the incomplete bleach of rhodopsin at photopic light intensities helps to preserve light responses in rods because a lower fraction of light activatable opsin molecules prevent response saturation.

In the current study, our aim was a near-total bleach of rhodopsin and the light intensity we used was more than three log units higher than the maximum intensity used by Munch and colleagues. Nevertheless, this is indeed a relevant citation with a more detailed and focused attention on this aspect. As suggested, we have now referenced this previous work as a relevant caveat in line 239 of the edited manuscript.

Point 4 iv)

Original Reviewer Comment 4iv): The authors also mention amacrine circuitry a number of times, but do not try and manipulate it. This would be interesting to explore to see if they can test how they generate the different ON and OFF responses, although this may be considered beyond the scope of the current study.

Author Response 4iv): Indeed, this would be something interesting for the future, particularly since synaptic re-wiring does occur in the degenerated retina. However, we also consider such investigations out of scope of the current manuscript.

New Reviewer Comment 4iv): We appreciate the authors response that this may be outside the scope of this study, however, it would be important to conduct these experiments to evaluate the efficacy of targeting bipolar cells with their optogene to know how the signal may spread laterally in the retina, especially as they discuss the lateral spread of native melanopsin responses in the dystrophic retina (native responses increasing from <10% to approximately 37% as stated by Eleftheriou et al., 2020). Might this be happening for the Mela MGLuR6 responses too? A brief consideration of this in the discussion would be sufficient.

New author Reply 4iv): We agree that the increased responsiveness of ipRGCs in the degenerated retina (treated and untreated) could hypothetically also affect the gating of optogene and bipolar cell mediated signaling to non-ipRGCs (see reply below to 4 vi) and that indeed, if this is happening in the *rd1* retina then this will also be the case for any Mela(CTmGluR6) or other optogenetic therapy expressed in a bipolar or amacrine cell of a *rd1* retina. Experimentally modulating inner retinal gating would indeed reveal if this mechanism enhances or decreases the efficacy of an ON bipolar cell targeted optogenetic therapy and a targeted modulation could potentially increase efficacy of treatment. We added two careful sentence eluding to above without speculations (line 414ff).

Point 4vi)

Original Reviewer Comment 4vi): Whilst this may be considered a personal preference, displaying data as a fraction of a whole graph is preferable over a pie chart, since the former allows one to better compare the proportional shifts between groups (WT vs *rd1* Mela(CTmGluR6), in this case). We would also suggest keeping the number of melanopsin cells in these comparisons, as we know that it should be consistent and not change between RGCs due to degeneration (approx. 6-8%).

Author Response 4vi): Thank you for the comment. In single-cell patch-clamp RGC recordings, melanopsin-like responses were ignored during an initial test stimulus. For this reason, melanopsin cells are not included in the patch-clamp recordings. The fraction of melanopsin responses are instead shown in the MEA recordings, where they were not excluded. Moreover, inclusion of the ipRGC may be mis-leading the data interpretation in this particular scenario, as there is indeed an increase of mel-anopsin-like responses in *rd1* mice, observed by us and others (Eleftheriou et al., 2020, doi: 10.3389/fnins.2020.00320.). This increase can climb to approx. 37% in the degenerated *rd1* retina.

New Reviewer Comment 4vi): We agree that the number of ipRGC light responses can increase in advanced degeneration through corrupt and functionally reorganised circuitry in the inner retina (as described by Eleftheriou et al., 2020 and others) but, to our knowledge, this has not been shown to be as a result of an increase in the number of melanopsin-expressing RGCs. This would however raise the question of whether the same method of widespread light activity in the inner retina (through electrical or chemical synapses) occurs when testing the efficacy of the Mela MGlur6 optogene in Bipolar cells? (as discussed in response 4iv). We appreciate that this is difficult to factor in for this graph but may be an important sentence to raise in the discussion until future experiments to address this are conducted.

New Author reply 4vi): An increase in the ipRGC responses is actually presented and factored in figure 6b and supplementary figure 10a. Suppl. Fig. 10a shows that a Mela(CTmGluR6)-treated rd1 retina displays about 37% of ipRGC-like responses, identical to the ipRGC response number in the non-treated rd1 retina found by Eleftheriou and colleagues. Fig. 6a shows a significantly higher percentage of responsive RGCs in treated rd1 retinas versus non-treated rd1 retinas, and this increase must therefore come from the optogenetic construct.

We are, however, also aware that ipRGCs feed back into the inner retinal network, for example by dopaminergic ACs (e.g. doi:10.1073/pnas.0803893105) and that enhanced ipRGC activity could potentially gate the transmission of the optogenetic signal from ON-Bipolar cells to RGCs, hypothetically also enhancing the response of other RGCs. However, such feedback mechanisms are yet entirely unexplored in the degenerated retina and therefore far beyond the scope of this paper. Based on existing physiological data and the current understanding of rewiring we feel that speculating on facilitation of inner retinal signaling will be arms waving. However, if indeed existent in the degenerating retina enhancing the efficacy of a bipolar cell targeted optogenetic therapy, this would be a very positive and exciting finding.

We added two careful sentence eluding to above without speculations (line 414ff).

Point 5. Figure 6a:

i) Original Reviewer Comment 5i): The % of responsive cells in the WT MEA recordings is surprisingly low at 60%. We would expect these to be more in the range of 90-100%. Can the authors comment on a potential explanation for this? The SD error bars also do not look like they should be NS compared to the treated mice. Perhaps a violin plot would be a better way of representing the true spread of the data points for all three groups?

Author Response 5i): To exclude any potential bias we have refrained from removing data from retinal pieces that did not show many light responsive cells, e.g. retinal pieces with little transduction. This is now also stated in the methods section. As suggested by the reviewer, we now added individual data points to the plots shown in Fig. 6a. One C57BL/6 retinal explant seemed to exhibit unusually weak light responses. We therefore confirmed normality of distributions using the Shapiro-Wilk test. We then performed a 2-tailed Student's T-test with Bonferroni correction ($\alpha=0.0166$), with the p-values stated in the manuscript. Additionally, we performed a one-way ANOVA (GraphPad, Prism, $\alpha=0.05$) that confirmed the results: C57BL/6 vs

rd1 adjusted p-value=0.003 (significant), C57BL/6 vs rd1_Mela(CTmGluR6) p= 0.1461 (not significant), rd1_Mela(CTmGluR6) vs rd1 p=0.0369 (significant).

New Reviewer Comment 5i): We appreciate the clarification and note the inclusion of the individual data points. This does allow the reader to now see the spread of the data. This does raise some further concern, though: 10% light responsive cells in a WT retina is clearly not typical i.e. it is much more likely to be a technical rather than biological basis. The majority of the recordings are up at >85%, which is what would also be expected from the published literature. Since the number of WT recordings is small (N = 6), the inclusion of that one data point brings the WT average down markedly. It also confers a very large SD to the WT data (almost twice that of the other two groups), which may be the reason that there is then no statistical difference compared to the Opto-treated RD1 cohort. The effect this has is to skew the data and give the impression that the Opto gene therapy is significantly better (i.e. closer to WT) than it actually is. It would seem likely that the removal of this one WT retina would remove the N.S. difference between WT and treated RD1.

We note that the decision to exclude data is a very difficult one and that the authors have used a normality test, but this is of limited strength on this sort of sample size. The inclusion of additional WT retina recordings to bring these control groups closer in N number to the treated numbers (N = 24) retinas would be an appropriate way to address the issue. This would allow a more accurate assessment of normality, as well as tests for outliers, and would most likely bring the SD within a range similar to that shown in the other groups.

New Author Reply 5i)

We have now performed additional WT recordings and boosted the N to 12. We also performed the ROUT method in graphpad to identify outliers and subsequently removed the lowest data point that was indeed identified as outlier. As predicted by the reviewer, the average RGC firing rate in the WT retina increased to >80% (Fig. 6a) and standard deviations are now similar across groups. We have additionally boosted the N of untreated rd1 retina recordings to 11. The new figure 6a also shows that C57BL/6 retinas possess higher firing rates compared to Mela(CTmGluR6)-treated *rd1* retinas, which was added to the new version of the manuscript (line 260ff).

Point 5 iii) Original Reviewer comment: The justification for use of a sigmoid curve to model the light- and dark-adapted intensity data points is unclear, as neither plot plateaus at a maximum and it is a significant assumption to state that they would. To describe a shift, the authors would need to reach maximum responses for both in order to fit the curves as they have done. The 'gold' data points look like they max out at 0.5 of the normalized response so it is unclear how they have plotted the curve shown to this particular data.

Author Response 5iii): Thank you very much for this attentive observation. The anticipated Hill fit was indeed performed incorrectly. In the revised version, the Hill fit was performed correctly on each individual curve, dark- and light-adapted. In figure 6c, both curves were normalized to Vmax retrieved from the Hill fit and plotted in the same graph. We have adapted the Methods section 4.5. by adding the following fit description: "Responses for each light intensity were normalized

to their minimal re-sponse in both, dark- and light-adapted trials and plotted as normalized averages \pm s.e.m. Half-saturations (EC50), Hill slope and maximal responses (Vmax) were calculated by fitting a Hill equation in GraphPad (Prism) to each individual curve. The values shown in the plots in Fig. 6c are normalized to Vmax." Hill fit slopes, Vmax and EC50 values are given in the legend of fig. 6c.

New Reviewer Comment 5iii):

We thank the authors for re-examining this, however we are still not certain that the revised presentation is correct mathematically. The graphs indicate that it was not possible to reach the maximum response in either condition, since neither plateaued, which is essential to accurately plot a sigmoidal dose-response curve regardless of whether the data is normalised or not. This would argue against the validity of trying to fit a sigmoid curve with an accurate hillslope. However, we do agree that there is a 1-unit log shift in the responses and so would recommend seeking the advice of a statistician in this instance, to see if there is a more appropriate fit.

New Author Reply 5iii): We have sought advice from a statistician (Dr. Beatriz Vidondo, Biostatistician and Data Scientist University of Bern, ORCID: 0000-0001-9693-2301; see attached Email below) and performed more experiments. In short, the used Hill equation is correct and neither the exponential nor logarithmic fit is congruent with the trend in our data.

Subject: Response to Reviewers

From: "Vidondo Curras, Beatriz Teresa (VETSUISSE)" <beatriz.vidondo@vetsuisse.unibe.ch>

Date: 11/08/2022, 12:09

To: Králik, Jakub (PYL) <jakub.kralik@unibe.ch>

CC: "Kleinlogel, Sonja (PYL)" <sonja.kleinlogel@unibe.ch>

Dear Jakub,

Dear Sonja,

many thanks for your request.

-the equation you used:

https://www.graphpad.com/guides/prism/latest/curve-fitting/reg_ecanything.htm

is correct; the data points fall nicely on the fitted lines.

-the whole trend in your data is congruent with the Hill equation for tissue responses according to the IUPHAR:

[https://en.wikipedia.org/wiki/Hill_equation_\(biochemistry\)#Tissue_response](https://en.wikipedia.org/wiki/Hill_equation_(biochemistry)#Tissue_response)

-the last two data points for both dark- and light-adapted mice show a change with respect to the previous trend

-the trend in your data is not congruent with neither exponential nor logarithmic equations

As a Biostatistician from the University of Bern, I approve the steps taken in this analysis. Best of luck with your current and future work.

Best regards,

Dr. Beatriz Vidondo

Biostatistician and Data Scientist

<https://www.researchgate.net/profile/Beatriz-Vidondo>

Veterinary Public Health Institute

University of Bern

www.vphi.ch